# A Machine Learning Method for the Fine-Grained Classification of Green Tea with Geographical Indication Using a MOS-Based Electronic Nose

**DOI:** 10.3390/foods10040795

**Published:** 2021-04-08

**Authors:** Dongbing Yu, Yu Gu

**Affiliations:** 1College of Information Science and Technology, Beijing University of Chemical Technology, Beijing 100029, China; yudongbing@mail.buct.edu.cn; 2Beijing Advanced Innovation Center for Soft Matter Science and Engineering, Beijing University of Chemical Technology, Beijing 100029, China; 3Guangdong Province Key Laboratory of Petrochemical Equipment Fault Diagnosis, Guangdong University of Petrochemical Technology, Maoming 525000, China; 4Department of Chemistry, Institute of Inorganic and Analytical Chemistry, Goethe-University, Max-von-Laue-Str. 9, 60438 Frankfurt, Germany

**Keywords:** green tea, electronic nose, convolutional neural network, support vector machine

## Abstract

Chinese green tea is known for its health-functional properties. There are many green tea categories, which have sub-categories with geographical indications (GTSGI). Several high-quality GTSGI planted in specific areas are labeled as famous GTSGI (FGTSGI) and are expensive. However, the subtle differences between the categories complicate the fine-grained classification of the GTSGI. This study proposes a novel framework consisting of a convolutional neural network backbone (CNN backbone) and a support vector machine classifier (SVM classifier), namely, CNN-SVM for the classification of Maofeng green tea categories (six sub-categories) and Maojian green tea categories (six sub-categories) using electronic nose data. A multi-channel input matrix was constructed for the CNN backbone to extract deep features from different sensor signals. An SVM classifier was employed to improve the classification performance due to its high discrimination ability for small sample sizes. The effectiveness of this framework was verified by comparing it with four other machine learning models (SVM, CNN-Shi, CNN-SVM-Shi, and CNN). The proposed framework had the best performance for classifying the GTSGI and identifying the FGTSGI. The high accuracy and strong robustness of the CNN-SVM show its potential for the fine-grained classification of multiple highly similar teas.

## 1. Introduction

Tea is one of the most popular drinks in the world and is enjoyed by many consumers. According to the reports of the Tea Association of the USA, $2.58 billion were spent in the specialty tea category in 2019, and green tea accounted for about 15% of the total tea consumption [1]. People appreciate green tea, a beverage with health-functional properties, for its benefits in preventing and controlling various diseases, such as different kinds of cancer, heart disease, and liver disease [2]. Green tea is a ‘non-fermented’ tea and typically consists of one tea bud covered in white hair and one or two leaves [3], with extremely small differences in aroma [4]; thus, it is not easy to distinguish green teas by appearance or aroma. The authentic green teas are processed using traditional methods, including fixation, rolling, and drying using baking or pan firing [5,6]. China has many green tea categories, including Longjing tea, Biluochun tea, Maofeng tea, and Maojian tea [7]. Each green tea category has several sub-categories with geographical indications (GTSGI), such as Xihu Longjing tea, Suzhou Biluochun tea, Huangshan Maofeng tea, and Xinyang Maojian tea, with significant price differences. Several high-quality GTSGI planted in specific areas are labeled as famous GTSGI (FGTSGI) and are very popular in China, such as Huangshan Maofeng tea in Anhui province and Xinyang Maojian tea in Henan province; these teas are among the ten famous teas in China [8,9]. The FGTSGI are typically more expensive than ordinary teas of the same sub-category, and fraudulent labeling of products has become a problem. Therefore, geographical origin discrimination is important to protect consumers’ interests and producers’ reputations. Typically, the discrimination of different teas is achieved by chemical detection [10] or sensory evaluation [11]. However, chemistry-based methods, such as gas chromatography-olfactometry [12] and gas chromatography-mass spectrometry [13], are time-consuming and laborious. Sensory evaluation methods rely heavily on taste experts, and the results may be affected by the individual’s physical condition or emotional state. Therefore, a rapid and convenient method is urgently needed to distinguish different teas objectively and reliably.

An electronic nose (E-nose) is a chemical measurement system that measures the chemical properties of sample gases and consists of a sampling system, a sensor array unit, and a data acquisition and processing system [14]. The sensor array can detect volatile compounds in the sample gases and provides feature information (sensor responses in the array). The E-nose has been increasingly used to analyze foods and beverages [15,16,17,18,19]. In recent years, many researchers have tried to distinguish tea categories using an E-nose system. Liu et al. [20] proposed a fast and effective random forest approach for distinguishing teas in six categories (white tea, green tea, black tea, yellow tea, dark tea, and oolong tea) with an E-nose and achieved an accuracy of 100%. Banerjee et al. [21] used an E-nose and a k-nearest neighbor algorithm to classify four black teas from different origins with an accuracy of 86.25%. Lu et al. [22] utilized an E-nose and a random forest approach to distinguish seven Longjing tea samples with different prices obtained from different companies and achieved an accuracy of over 95%. Although the E-nose has been successfully applied for analyzing teas, most studies primarily focused on distinguishing teas in different categories [20,23,24] or teas in the same category from different production areas or with different qualities [21,22,25]. Few studies focused on the fine-grained classification of tea in different categories (categories) and from different production areas (sub-categories) simultaneously. Green teas typically have light aromas with subtle differences between different green tea categories or sub-categories. These slight differences coupled with an increasing number of sub-categories have complicated the classification. Therefore, the evaluation of the discriminative characteristics that distinguish different teas is not straightforward. Many studies focused on improving the performance of the E-nose system by selecting sensitive material [26], optimizing the sensor array [27], or enhancing the feature extraction methods and algorithms [28,29]. The feature extraction method is a crucial aspect of the optimization methods, and it is important to extract useful information from the sensor signals [30] To the best of our knowledge, most researchers employing E-noses used steady-state or transient-state features (maximum values [31,32], stable values [33,34], integrals [35,36], and derivatives [36,37]). However, using only one feature may miss crucial information useful for classification, and manual fusion of multiple features is challenging due to differences in the sensor types and the detection targets in real applications.

A convolutional neural network (CNN) is a feedforward neural network designed to process data in multiple arrays by extracting local and global features [38]. CNNs can automatically mine the potential information using an appropriate convolution structure and feed the information to a classifier providing good performance. Due to their strong feature extraction ability and excellent performance, CNNs are widely used for image processing [39], speech recognition [40], and processing matrix data. A CNN is a deep learning method, and large amounts of data are essential for model training, which may prove difficult in practical applications. Therefore, selecting a suitable classifier is vital. Due to the limitations of CNNs, few studies used them in E-nose applications. In 2019, Shi et al. [41] employed a CNN model to extract the volatile compound profile of beer using an E-nose. Ten metal oxide semiconductor (MOS) sensors were used, and the first 90 sample points obtained from each sensor (90 × 10 sample points) were converted into a 30 × 30 matrix for input to the CNN. Although the proposed method achieved a classification accuracy of 96.67%, the 30 × 30 input matrix (transformed simply from 10 sensors’ data) was difficult to interpret. The sensor array in an E-nose system has high cross-sensitivity because each sensor in the E-nose is typically not sensitive for only a single kind of substance, but rather for a certain range of substances, some of which do overlap [42]. Thus, it is essential to consider the correlation between the sensors to improve classification accuracy.

This paper presents an effective approach for the fine-grained classification of green tea with geographical indication based on a novel deep learning framework using a MOS-based E-nose. The framework consists of a CNN backbone and a support vector machine (SVM) classifier and is referred to as CNN-SVM. This study makes the following contributions:A preprocessing scheme inspired by the multi-channel input to the CNN model used in image processing is proposed for the 10-channel E-nose’s data. The channel’s sensor data are converted into a single matrix, and the combined matrices of the 10 channels represent the multi-channel data.The network structure of the proposed CNN-SVM, a CNN backbone with a SVM classifier, is analyzed. In the framework, the deep CNN is designed to mine the volatile compound information of the tea samples (automatically), and the SVM is used to classify the data (small sample sizes) to improve the classification performance.A comprehensive study on the fine-grained classification of green tea with geographical indication is presented, demonstrating the high accuracy and strong robustness of the proposed CNN-SVM framework.

## 2. Materials and Methods

### 2.1. General Information of Tea Samples

Twelve green teas (six Maofeng teas and six Maojian teas) from different geographical origins were used in this study, including Huangshan Maofeng (HSMF), Yandang Maofeng (YDMF), Emei Maofeng (EMMF), Lanxi Maofeng (LXMF), Qishan Maofeng (QSMF), Meitan Maofeng (MTMF), Xinyang Maojian (XYMJ), Guilin Maojian (GLMJ), Duyun Maojian (DYMJ), Guzhang Maojian (GZMJ), Queshe Maojian (QSMJ), and Ziyang Maojian (ZYMJ). The details of the producing areas are presented in Table 1. The 12 green teas came from eight provinces (Anhui, Zhejiang, Sichuan, Guizhou, Henan, Guangxi, Hunan, and Shaanxi) in China; the geographical information is shown in Figure 1.

Among these green teas, HSMF tea and XYMJ tea are the two most important products with a protected designation of origin in China. HSMF tea is a famous Maofeng green tea produced in Huangshan City, Anhui Province (29°42${}^{\prime }$33.26${}^{\prime \prime }$ N, 118°19${}^{\prime }$02.37${}^{\prime \prime }$ E); the protection zone of the HSMF tea is shown in Figure 2a [43]. The tea is primarily grown near Huangshan Mountain at altitudes of over 700 m [8]. The annual rainfall in this area is 1500–1800 mm, the average annual temperature is 15.5–16.4 °C, and the cumulative sunshine hours are 1674–1876 h. The humid, cloudy conditions are an ideal growing environment for high-quality tea. XYMJ tea is a well-known local product in Xinyang City, Henan Province (32°07${}^{\prime }$23.79${}^{\prime \prime }$ N, 114°04${}^{\prime }$30.11${}^{\prime \prime }$ E). The protection zone of the XYMJ tea is shown in Figure 2b [44]. The tea plantations are located on the western part of the north slope of Dabie Mountain and the eastern part of the north slope of Tongbai Mountain. The average altitude is 500–800 m. The annual rainfall in this area is 900–1400 mm, the average annual temperature is 15.1–15.3 °C, and the cumulative sunshine hours are 1900–2100 h. XYMJ tea grown in this area has a light fragrance and has a sweet, heavy, and mellow taste [45].

### 2.2. Experimental Samples and Conditions

In this study, all the tea samples (twelve teas with special-class levels) were purchased from specialty stores approved by the China Tea Marketing Association. The tea was picked and processed 14 days before the Pure Brightness Festival (around April 5 or 6) in 2019 and is called “before-brightness tea”. The details of the tea samples are listed in Table 1. Photos of the 12 green teas are shown in Figure 3.

A commercial E-nose (PEN3, Airsense Analytics GmbH, Germany; the details of the PEN3 can be found on the company’s website [46]), with 10 MOS sensors was used to acquire volatile compound profile of the tea samples. The details of the MOS sensors are shown in Table 2 [47].

The experiments were conducted for 12 days in a clean testing room of the authors’ laboratory (with good ventilation and an area of about 45 square meters) at a temperature of 25 °C ± 1 °C and a humidity level of 40% ± 2%. The 12 green teas (six Maofeng teas and six Maojian teas) were measured 10 times per day by a professional and experienced operator, and the tea samples for daily experiments were updated. The acquisition of the volatile compound profile was conducted in a well-ventilated location to minimize baseline fluctuations and interference from other volatile compounds. The zero gas (a baseline) was produced using two active charcoal filters (Filter 1 and Filter 2 in Figure 4) to ensure that the reference air and the air used for the samples had the same source.

The workflow of the E-nose includes the collection stage and flushing stage. As is shown in Figure 4, for each tea, a 4 g tea sample was placed into a 50 mL sampler for 180 s to allow the tea’s volatile compounds to disperse into the sampler. Before the measurement, clean air was pumped through filter 2 into the E-nose with a flow rate of 10 mL/s in the direction of In 2 for 100 s. The automatic adjustment and calibration of the zero gas is called zero-point trim; the values relative to the zero-point values were recorded as a baseline. After the calibration, the tea sample’s volatile gas in the sampler was pumped into the E-nose with a flow rate of 10 mL/s in the direction of In 1 to contact the sensor array for 100 s. The gas molecules were adsorbed on the sensors’ surface, changing the sensors’ conductivity due to the redox reaction on the surface of the sensor’s active element. The sensors’ conductivity eventually stabilized at a constant value when the adsorption was saturated. The collection stage lasted 100 s, and sampling continued at one sample per second. Figure 5 shows examples of the 10 sensors’ response curves in the collection stage for the 12 tea samples. The response value *R* of the sensor was calculated with the equation $R=G_{0}/G$, where $G_{0}$ is the conductivity of the sensor in reference air (in the calibration stage), and *G* is the conductivity of the sensor exposed to the sample vapor (in the data collection stage). The sensor chamber was flushed with the zero gas between measurements. In this step, the clean air was pumped into the E-nose with a flow rate of 10 mL/s in the direction of In 2 for 100 s. The zero gas flushed the sensor surface to completely remove the analytes. The flushing and data collection stages were repeated to obtain the raw data of 12 tea samples. Therefore, the dataset contained 1440 measurements (12 tea samples × 10 measuring times × 12 days).

### 2.3. Principal Component Analysis (PCA)

A PCA is a multivariate statistical technique used to reduce the dimensionality of raw complex data and extract meaningful information [48]. With minimal effort, PCA can be used to construct a set of new orthogonal variables called principal components using orthogonal linear transformation [49]. The first few principal components often have large variances and explain most of the sample variance. Usually, the first few components, whose cumulative variance exceeds 95% are selected as the principal components [50]. Thus, the PCA is an effective data compression method. Similar data points tend to be clustered and unrelated data points are scattered in the PCA projection plot.

### 2.4. Convolutional Neural Network (CNN)

A CNN is a feedforward neural network designed to process data in multiple arrays by extracting local and global features [38]. In image processing, the CNN input is typically a color image consisting of three 2D arrays containing pixel information in three channels. Due to the flexibility of the data modalities, a CNN can be used in many fields, such as natural language processing, remote sensing, and sensory data analysis. A typical architecture of a CNN consists of an input layer, a hidden layer, and an output layer. The hidden layer of a CNN is composed of a convolution layer, a pooling layer, and a fully-connected layer. The convolution layer and pooling layer are stacked; the former detects local features, and the latter merges semantically similar features into one to obtain the high-level features of the input data. The fully-connected layer is used as a classifier. Each node is connected with all the upper layer nodes to map the feature information to the decision space [41]. The CNN provides high classification accuracy due to its unique structure.

### 2.5. ResNeXt

ResNeXt is a novel, simple, and highly modularized network based on the CNN and is designed to improve the image classification accuracy due to its homogeneous, multi-branched architecture [51]. Unlike existing CNN-based networks, such as VGGnet [52] and ResNet [53], the ResNeXt model has the cardinality dimension in addition to width and depth. Cardinality is defined as the size of the set of transformations that are aggregated to construct a network with the same topology. The addition of cardinality proved to be more effective than increasing the depth and width, especially when the latter leads to diminishing returns. Thus, the ResNeXt can achieve high accuracy with low complexity for visual (and non-visual) recognition tasks.

### 2.6. Support Vector Machine (SVM)

An SVM is a supervised learning and classification algorithm based on statistical theory [54]. An SVM can map input data that is not linearly separable in a low-dimensional space to a high-dimensional space non-linearly using a kernel function. A hyperplane is constructed in the high-dimensional space to maximize the margin between two classes and classify the data in the high-dimensional space. The most commonly used kernel function is the radial basis function (RBF), which has resulted in high classification performance [55]. Since SVM performs structural risk minimization, it is regarded as a good classifier for nonlinear data and small sample sizes.

### 2.7. Model Evaluation Metrics

Model evaluation metrics are used to assess the algorithm performance in supervised learning. Common metrics include accuracy, recall, precision, F1 score, and Kappa score. As shown in Figure 6, a confusion matrix that provides the True Positive (TP), False Positive (FP), True Negative (TN), and False Negative (FN) parameters is used to calculate the metrics.

Accuracy represents the proportion of samples that are correctly classified (including TP samples and TN); it is defined as follows:(1)$$Accuracy=\frac{TP+TN}{TP+TN+FP+FN}$$

Recall(R) is defined as the ratio of the TP samples to the sum of the TP and FN samples. Precision(P) is defined as the ratio of the TP samples to the sum of the TP and FP samples. The equations of recall and precision are:(2)$$Recall=\frac{TP}{TP+FN}$$
(3)$$Precision=\frac{TP}{TP+FP}$$

The F1 score considers the recall and precision and is calculated as follows:(4)$$F1=\frac{2PR}{P+R}$$

The Kappa score is a statistic that measures the agreement between categories and is used to evaluate the classification accuracy; it is calculated as:(5)$$p_{o}=\frac{TP+TN}{TP+TN+FP+FN}$$
(6)$$p_{e}=\frac{\left(TP+FN\right)\left(TP+FP\right)\left(TN+FN\right)\left(TN+FP\right)}{{\left(TP+TN+FP+FN\right)}^{2}}$$
(7)$$Kappa=\frac{p_{o}-p_{e}}{1-p_{e}}$$
where $p_{o}$ represents the total classification accuracy, and $p_{e}$ represents the expected agreements.

## 3. Proposed Method

A CNN with an appropriate structure can mine the deep features of the input data and has a flexible form. However, more data are required for CNN model training than for a traditional statistical model [41], and the data volume may not be sufficient for practical E-nose applications, potentially resulting in overfitting. The SVM is considered a good classifier for small sample sizes, but it is difficult to select or extract valid features using the SVM. In this study, we combined the advantages of these two algorithms and proposed the CNN-SVM framework. The deep CNN structure was used to mine the volatile compound information of the tea from the raw data of the E-nose, and the SVM was used to classify the data to achieve high classification accuracy. The proposed CNN-SVM framework consisting of a CNN backbone and an SVM classifier is shown in Figure 7.

### 3.1. Data Preprocessing

The 10-channel sensor array in the PEN3 system has high cross-sensitivity because each sensor is sensitive to a certain range of substances, and there is some overlap. The E-nose data need to be preprocessed before determining the correlation between the sensors for the input of the CNN model. Therefore, data preprocessing is crucial to improve the classification accuracy. Inspired by the multi-channel input to the CNN model used in image processing, the raw data of the 10 sensors were converted into a 10-channel input for the CNN. As is shown in Figure 7, the matrix form of the original data was 100 × 10, where 100 represents the number of sampling times in the data collection stage (100 s) for each sensor, and 10 represents the number of sensors. The 100 raw data points obtained from each sensor were converted into a 10 × 10 matrix, and the 10 matrices of the sensors were concatenated into a 10 × 10 × 10 matrix as a 10-channel input of the CNN backbone.

### 3.2. Proposed CNN-SVM Framework

Inspired by ResNeXt, a CNN backbone was designed to extract the deep features from the 10-channel input data. As shown in Figure 8, the CNN backbone consisted of four stages, including Conv 1, Layer 1, Layer 2, and Pool 1. The Conv 1 stage had a kernel size of 1 × 1 and a stride of 1 to expand the number of input channels when a large number of feature maps were constructed with features from different channels. The Layer 1 stage and Layer 2 stage, which consisted of m ResBlocks and n ResBlocks, respectively, were used to reduce the size of the feature maps when a large number of deep features were obtained (m = 3 and n = 3 in this study). As shown in Figure 9, the input feature maps were processed in the ResBlock by several grouped convolutional layers (the first 1 × 1 layer, the 3 × 3 layer, and the second 1 × 1 layer). The number of grouped convolutional layers was defined as cardinality (c) and was set to 4 in this study. In each grouped convolution, the first 1 × 1 layer had a kernel size of 1 × 1 and a stride of 1 to adjust the number of input channel (in_ch) to bottleneck width (bw). The 3 × 3 layer in the grouped convolutional layers had a 3 × 3 convolution kernels with a stride of s, whose input and output channels were both bw-dimensional (bw = 4 in this study) to obtain the valid features of the feature maps. The number of output channels (out_ch) was adjusted by the second 1 × 1 layer with a kernel of 1 × 1 and a stride of 1, and the outputs of the grouped convolutional layers were concatenated along with the output of the input by a skip connection as the output of the ResBlock. The skip connection retained features in different layers in the forward propagation that allowed the model to reuse these features and accelerate the training speed. In this study, the in_ch was 16, the s was 1, and the out_ch was 32 in Layer 1 for each ReBlock, respectively. Thus, the sizes of the input feature maps remained the same when more deep features were obtained. In Layer 2, the in_ch was 32, and the out_ch was 64 for each ResBlock. The s of the first ResBlock was 2, and the s of the two other ResBlocks was 1 to reduce the sizes of the feature maps when local features were obtained. Sixty-four features were obtained by the global average layer, and the feature map of each channel was integrated into one feature. The details of the CNN backbone are listed in Table 3.

The 64 features obtained by the CNN backbone were first fed to the CNN classifier consisting of two fully-connected layers for the training of the CNN backbone. The first fully-connected layer was activated by a ReLU function with 32 neurons, and the second fully-connected layer was activated by a Sigmoid function with 12 neurons to increase the nonlinear expression ability of the model. A weighted cross-entropy loss function was used and was minimized by the stochastic gradient descent (SGD) optimizer with a learning rate of a (a = 0.001 in this study). A warm-up strategy was used in the initialization of the learning rate for a smooth training start, and a reduction factor of b (b = 0.0001 in this study) was used to reduce the learning rate after every training epoch.

The batch size was 12, and the CNN framework (CNN backbone + CNN classifier) was trained for d (d = 500 in this study) epochs. When the model had converged on the training set, the model parameters of the CNN backbone were saved. Subsequently, the CNN classifier was replaced by the SVM classifier in the CNN backbone to construct the CNN-SVM model. The RBF was chosen as the kernel function, and a grid search method was used to search the other two important parameters (penalty factor and gamma) to obtain the best performance of the SVM model. Ten-fold cross-validation was used for training the SVM model; the training set was the same as the one used for CNN training. The proposed CNN-SVM algorithm is summarized in Algorithm 1.
**Algorithm 1** CNN-SVM**Input:** training data.**Output:** predicted category.**Begin**Step1: train the CNN backbone with the CNN classifier on the training dataStep2: save the parameters of the CNN backboneStep3: replace the CNN classifier with the SVM classifierStep4: test the CNN-SVM model on the test dataStep5: output the classification results**End**

## 4. Results and Discussion

### 4.1. Principal Component Analysis

PCA is an unsupervised method commonly used for pattern recognition. PCA can intuitively show the data distribution in a low-dimensional space by extracting meaningful information from the raw data. Since the conventional method uses a stable value as the static characteristic, the data obtained from each sensor from 81–100 s from 120 randomly chosen tea samples (12 tea samples × 10 measuring times × 1 day) was used as an example to present the PCA result. The projections of the two first primary components of the PCA computed for the tea samples is shown in Figure 10. The x-axis represents principal component 1 (PC 1) with 70.92% variance, and the y-axis represents principal component 2 (PC 2) with 24.51% variance. The cumulative variance of PC 1 and PC 2 was 95.43%, showing a small loss of information. The PCA method allows data to be grouped according to the similarity of the input characteristics to determine the distance between classes as shown in Figure 10. If the clusters are well separated, high classification accuracy is expected. If the clusters are in close proximity or overlap, low accuracy is expected. As shown in Figure 10, the PCA results show a relatively high overlap between the classes, indicating that this method is not suitable for separating the classes.

### 4.2. Comparison of the Classification Results of five Models

In this study, the classification performances of the SVM and CNN used in the CNN-SVM were compared with that of the proposed CNN-SVM model using multiple evaluation metrics (accuracy, recall, precision, F1 score, and Kappa score). The models were implemented using two NVIDIA GTX 1080TI graphics cards and the open-source PyTorch framework.

As a conventional method that uses the static characteristics of the gas sensor signals, the SVM model was first used to distinguish the 12 teas from different areas. Similar to the PCA approach, the last 20 data points (81–100 s) of the 10 sensors were used as input to the SVM model. A dataset of 1080 measurements obtained in the first nine days (12 tea samples × 10 measuring times × 9 days) was used for training and was expressed as a 21,600 × 10 matrix. A dataset of 360 measurements obtained in the last three days (12 tea samples × 10 measuring times × 3 days) was used for testing and was expressed as a 7200 × 10 matrix. The RBF was chosen as the kernel function of the SVM model. A grid search method was used to search the two important parameters with the best performance using 10-fold cross-validation on the training set. The parameters included the penalty factor [1, 10, 50, 100, 200, 500] and the gamma [0.01, 0.1, 1, 5, 10, 20]. A training accuracy of 100% was obtained. Thus, the penalty factor and the gamma were set to 100 and 1, respectively. The classification results of the SVM model on the test set are shown in Table 4. Five evaluation metrics, including accuracy, recall, precision, F1 score, and Kappa score, were computed for a comprehensive and accurate assessment of the classification model. The SVM achieved an accuracy of 72.5%, a recall of 72.5%, a precision of 86.24%, an F1 score of 75.24%, and a Kappa score of 70%, respectively. The SVM model provided unsatisfactory performance, and the experimental results showed that it lacked generalization ability and did not mine a sufficient number of features useful for the fine-grained classification of 12 green teas from different geographical origins using only the stable values.

The proposed CNN framework consisting of the CNN backbone and the CNN classifier was given the 10-channel input and trained using the CNN backbone. The dataset of 1080 measurements in the first nine days (12 tea samples × 10 measuring times × 9 days) was used for training and was expressed as a 1080 × 10 × 10 matrix. The dataset of 360 measurements in the last three days (12 tea samples × 10 measuring times × 3 days) was used for testing and was expressed as a 360 × 10 × 10 matrix. For training the CNN model, 10-fold cross-validation was used to verify the model performance on the training set. The loss and cross-validation curves of the CNN model during the training process are shown in Figure 11. The CNN model converged after 480 epochs. We saved the model parameters and tested the CNN model on the test set. As shown in Table 4, the CNN model achieved relatively good performance with an accuracy of 91.39%, a recall of 91.39%, a precision of 93.75%, an F1 score of 91.52%, and a Kappa score of 90.61%. The experimental results showed that the deep features extracted from the 10-channel input, which considered the correlation between the sensors, provided better performance than the SVM model for the fine-grained classification of the 12 green teas.

The proposed CNN-SVM framework consisting of the CNN backbone and the SVM classifier was provided with the 10-channel input. The training set and test set were the same as those used for the CNN framework. The CNN backbone in the CNN-SVM framework used the model parameters of the trained CNN framework at 480 epochs. A grid search method was used to search the penalty factor [1, 10, 50, 100, 200, 500] and the gamma [0.01, 0.1, 1, 5, 10, 20] with the best performance using 10-fold cross-validation on the training set. The penalty factor was set to 1, and gamma was set to 0.1; a training accuracy of 100% was achieved on the training set. As shown in Table 4, the CNN-SVM model achieved the best performance on the test set, with an accuracy of 96.11%, a recall of 96.11%, a precision of 96.86%, an F1 score of 96.03%, and a Kappa score of 95.76%. The experimental results indicated that the SVM classifier significantly improved the classification performance due to its good discrimination ability for small sample sizes. The effectiveness of the combination of the CNN and SVM was demonstrated for the classification of tea using an E-nose system in a practical application.

Furthermore, the CNN model proposed by Shi et al. [41] (CNN-Shi) was also tested for comparison with the proposed method. For consistency, in the CNN-Shi model, the first 90 sampling points for each sensor (90 × 10 totally) were converted into a 30 × 30 matrix for the CNN. The CNN-Shi framework consisted of 4 convolutional layers, 3 pooling layers, and 2 fully-connection layers. After the convolutional and pooling operations, 128 features were obtained to as input to the fully-connected layers to obtain the classification results. The training set and test set were consistent with those used in the proposed CNN framework. The CNN-Shi model was also trained for 500 epochs and achieved an accuracy of 85.27%, a recall of 85.27%, a precision of 86.83%, an F1 score of 85.16%, and a Kappa score of 89.39%, as shown in Table 4. In addition, we also replaced the fully-connected layers of the CNN-Shi model with an SVM to establish the CNN-SVM-Shi model for comparison with the proposed CNN-SVM model. The model parameters of the CNN-Shi model were saved at 480 epochs, and a grid search method was also used to search the best parameters for the SVM. The results are shown in Table 4. The model had an accuracy of 91.39%, a recall of 91.39%, a precision of 93.75%, an F1 score of 91.52%, and a Kappa score of 90.61%. The performances of the proposed CNN model and CNN-SVM model are better than those of the CNN-Shi model and CNN-SVM-Shi model, respectively. The experimental results further demonstrated the effectiveness of the 10-channel input of the CNN model due to the cross-sensitivity of the sensors. The results indicate that the proposed method provided better performances for mining deep features from the sensor signals and conduct a fine-grained classification using the E-nose system.

An experiment was performed (distinguishing the two FGTSGI from the 12 GTSGI) to demonstrate the practical applicability of the CNN-SVM framework. The F1-scores of the classes are listed in Table 5. The proposed CNN-SVM model outperformed the other four models for the identification of the two most famous and expensive teas with protected designation of origin (F1 scores of 97.77% for HSMF and 99.31% for XYMJ). The experimental results indicated that the model could accurately distinguish HSMF and XYMJ from the other Maofeng and Maojian tea sub-categories.

## 5. Conclusions

In this study, a CNN-SVM framework was proposed for the fine-grained classification of green tea with geographical indication using a MOS-based E-nose. The experiments were conducted on 12 green tea samples (six Maofeng green teas and six Maojian green teas). The main conclusions of this study are as follows:PCA, an unsupervised method, was used to show the separability of the 12 tea sample data (stable values) obtained from the E-nose. Not surprisingly, the PCA results showed that this method was ineffective for classifying the sample due to relatively high overlap. The stable values combined with an SVM model were used to distinguish the teas; this approach had poor performance. It showed that the static characteristics were not sufficient for the fine-grained classification of the 12 green teas, and meaningful feature information was lost.A 10-channel input matrix, which was obtained by converting the raw data of the 10-channel MOS-based sensors (each sensor for one channel) in the E-nose system, was constructed to mine the deep features using the CNN model. The multi-channel design considered the cross-sensitivity of the sensors and contained sufficient details for the classification tasks, providing a novel feature extraction method for E-nose signals in practical applications.A CNN-SVM framework consisting of the CNN backbone and the SVM classifier was proposed to perform the classification using the 10-channel input matrix. The novel structure of the CNN backbone based on ResNeXt was effective for extracting the deep features from the different channels automatically. The SVM classifier improved the generalization ability of the CNN model and increased the classification accuracy from 91.39% (CNN backbone + CNN classifier) to 96.11% (CNN backbone + SVM classifier) among the 12 green teas due to its good discrimination ability for small sample sizes.Compared with the other four machine learning models (SVM, CNN-Shi, CNN-SVM-Shi, and CNN), the proposed CNN-SVM provided the highest scores of the five evaluation metrics for the classification of the GTSGI: accuracy of 96.11%, recall of 96.11%, precision of 96.86%, F1 score of 96.03%, and Kappa score of 95.76%. Excellent performance was obtained for identifying the FGTSGI, with the highest F1 scores of 97.77% (for HSMF) and 99.31% (for XYMJ). These experimental results demonstrated the effectiveness of the CNN-SVM for the classification of the GTSGI and the identification of the FGTSGI.

In conclusion, the combined strategy of CNN and SVM enhanced the detection performance of multiple highly similar green teas. The proposed method provided high classification accuracy, showing that the tea quality differed for different geographical indications. Moreover, the method is rapid, convenient, and effective for classifying green teas. In the future, the potential of this method will be explored for other teas (or other foods) to expand the application of the proposed framework combined with the E-nose. We expect that the framework has a promising potential for using machine learning methods for food authentication.

## Figures and Tables

**Figure 1 foods-10-00795-f001:**
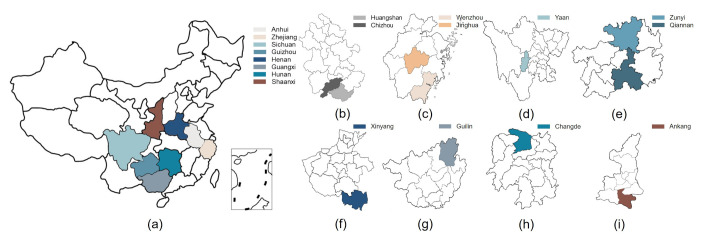
Geographical information of the tea-producing areas. (**a**) Location of the eight provinces (Anhui, Zhejiang, Sichuan, Guizhou, Henan, Guangxi, Hunan, and Shaanxi) in China; (**b**) Huangshan City and Chizhou City in Anhui Province; (**c**) Wenzhou City and Jinghua City in Zhejiang Province; (**d**) Yaan City in Sichuan Province; (**e**) Zunyi City and Qiannan Prefecture in Guizhou Province; (**f**) Xinyang City in Henan Province; (**g**) Guilin City in Guangxi Province; (**h**) Changde City in Hunan Province; (**i**) Ankang City in Shaanxi Province.

**Figure 2 foods-10-00795-f002:**
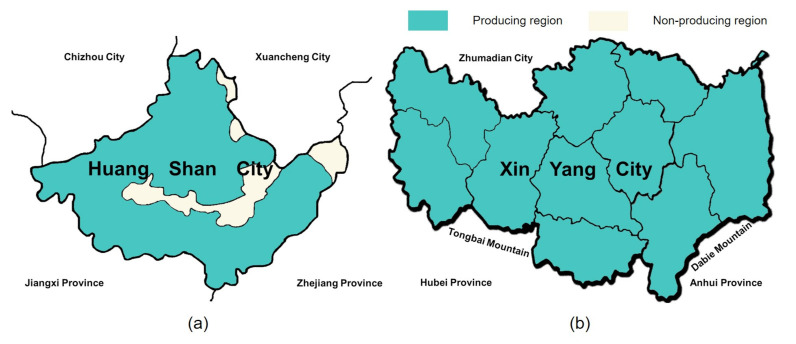
Protection zones for products of geographical indication. (**a**) HSMF tea in Huangshan City, Anhui Province; (**b**) XYMJ tea in Xinyang City, Henan Province.

**Figure 3 foods-10-00795-f003:**
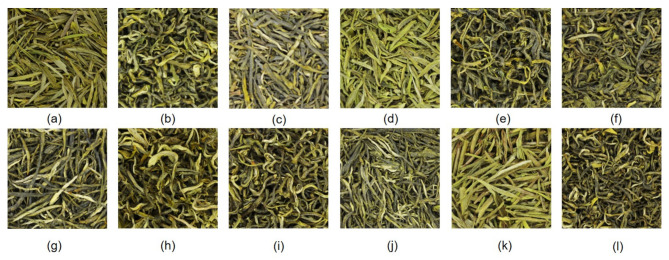
Photos of the 12 green teas. (**a**) HSMF; (**b**) YDMF; (**c**) EMMF; (**d**) LXMF; (**e**) QSMF; (**f**) MTMF; (**g**) XYMJ; (**h**) GLMJ; (**i**) DYMJ; (**j**) GZMJ; (**k**) QSMJ; (**l**) ZYMJ.

**Figure 4 foods-10-00795-f004:**
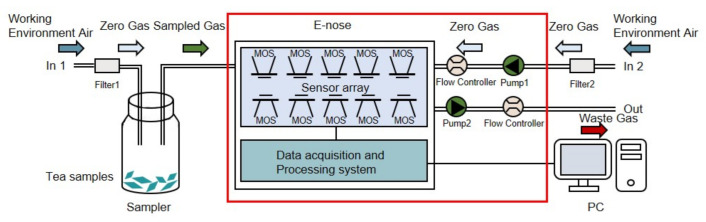
Schematic diagram of the PEN-3 workflow.

**Figure 5 foods-10-00795-f005:**
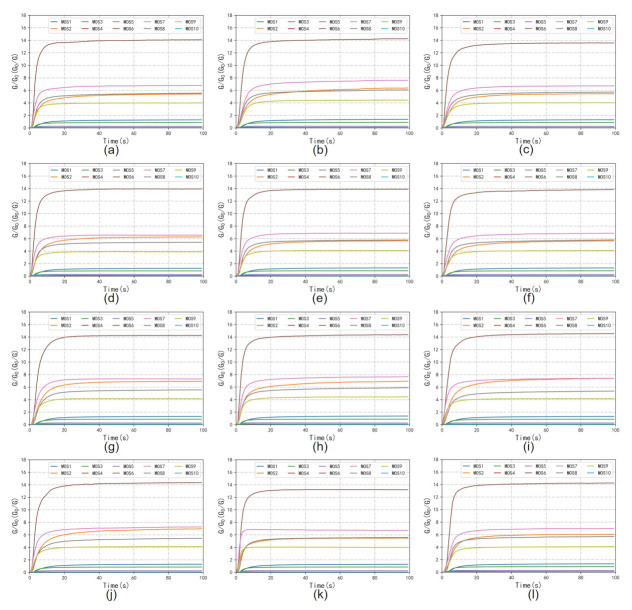
Examples of the 10 sensors’ response curves. (**a**) HSMF; (**b**) YDMF; (**c**) EMMF; (**d**) LXMF; (**e**) QSMF; (**f**) MTMF; (**g**) XYMJ; (**h**) GLMJ; (**i**) DYMJ; (**j**) GZMJ; (**k**) QSMJ; (**l**) ZYMJ.

**Figure 6 foods-10-00795-f006:**
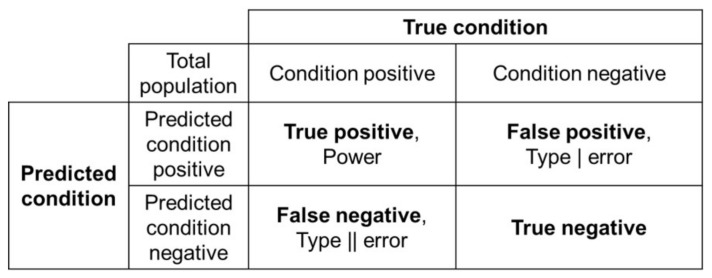
Calculation of the performance metrices by confusion matrix.

**Figure 7 foods-10-00795-f007:**
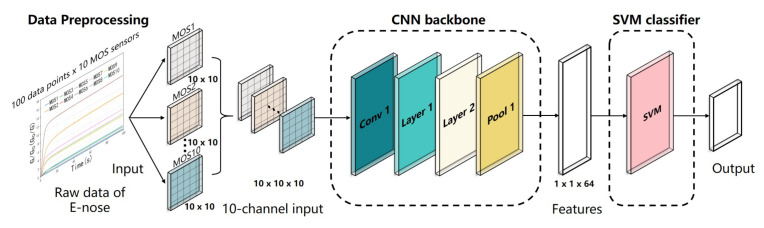
Schematic diagram of the CNN-SVM structure.

**Figure 8 foods-10-00795-f008:**
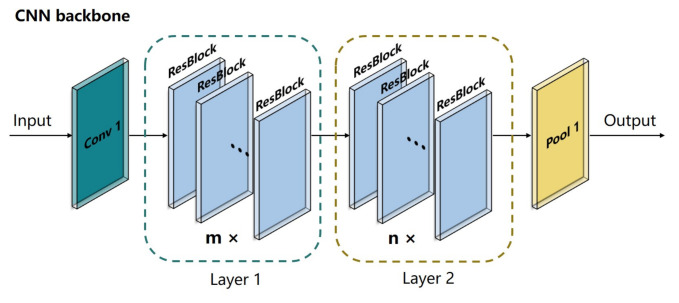
Schematic diagram of the CNN backbone.

**Figure 9 foods-10-00795-f009:**
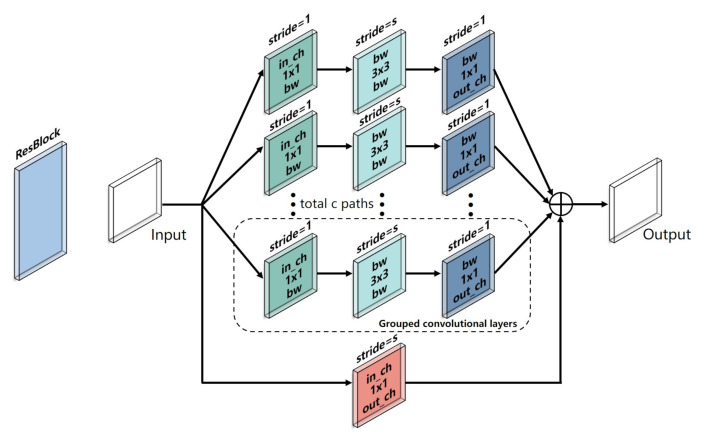
Schematic diagram of the ResBlock.

**Figure 10 foods-10-00795-f010:**
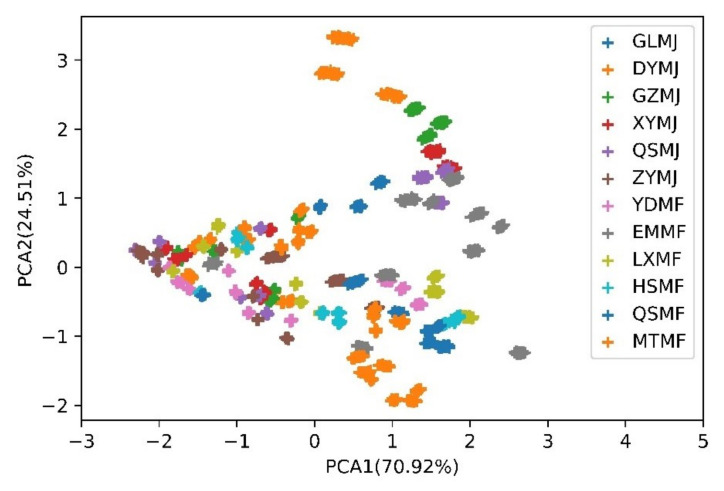
Projection of the first two principal components of the PCA of the tea samples.

**Figure 11 foods-10-00795-f011:**
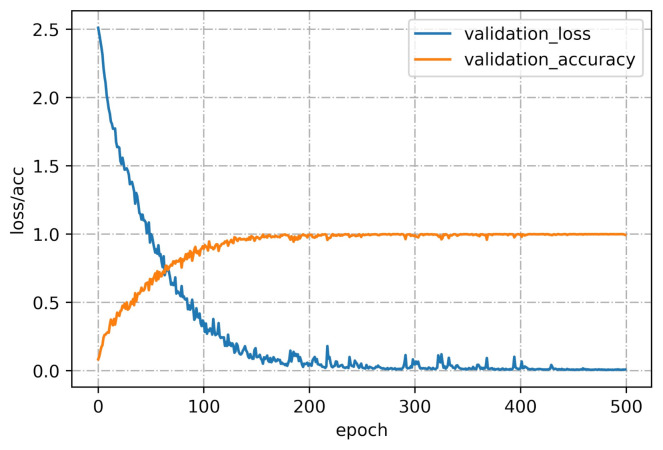
Loss and cross-validation curves of the CNN model during the training process.

**Table 1 foods-10-00795-t001:** Details of the 12 tea samples.

No.	Maofeng Tea			Maojian Tea		
	Name	Producing Area	Price ($/50 g)	Name	Producing Area	Price ($/50 g)
1	HSMF	Huangshan City, Anhui Province	47.2	XYMJ	Xinyang City, Henan Province	42.1
2	YDMF	Wenzhou City, Zhejiang Province	39.4	GLMJ	Guilin City, Guangxi Province	17.7
3	EMMF	Yaan City, Sichuan Province	15.5	DYMJ	Qiannan Prefecture, Guizhou Province	39.8
4	LXMF	Jinghua City, Zhejiang Province	26.0	GZMJ	Changde City, Hunan Province	35.5
5	QSMF	Chizhou City, Anhui Province	22.6	QSMJ	Zunyi City, Guizhou Province	16.8
6	MTMF	Zunyi City, Guizhou Province	13.1	ZYMJ	Ankang City, Shaanxi Province	27.3

**Table 2 foods-10-00795-t002:** Details of the 10 sensors in the PEN3.

No.	Sensor	Main Performance
1	W1C	Sensitive to aromatic compounds
2	W5S	High sensitivity to nitrogen oxides, broad range sensitivity
3	W3C	Sensitive to ammonia and aromatic compounds
4	W6S	Sensitive mainly to hydrogen
5	W5C	Sensitive to alkanes and aromatic components and less sensitive to polar compounds
6	W1S	Sensitive to methane, broad range sensitivity
7	W1W	Sensitive primarily to sulfur compounds and many terpenes and organic sulfur compounds
8	W2S	Sensitive to ethanol and less sensitive to aromatic compounds
9	W2W	Sensitive to aromatic compounds and organic sulfur compounds
10	W3S	Highly sensitive to alkanes

**Table 3 foods-10-00795-t003:** Details of the CNN backbone.

Stage	Output	Structure Details
Conv 1	10 × 10 × 16	1 × 1, 16, *stride*=1
Layer 1	10 × 10 × 32	$\left[\begin{array}{c} 1\times 1,\;16\\ 3\times 3,\;16\\ 1\times 1,\;32 \end{array}\right]\times 3$
Layer 2	5 × 5 × 64	$\left[\begin{array}{c} 1\times 1,\;32\\ 3\times 3,\;32\\ 1\times 1,\;64 \end{array}\right]\times 3$
Pool 1	1 × 1 × 64	global average pool

**Table 4 foods-10-00795-t004:** Results of five evaluation metrics for classifications by five models.

Model	Accuracy	Recall	Precision	F1 Score	Kappa Score
SVM	0.7250	0.7250	0.8624	0.7524	0.7000
CNN-Shi [41]	0.8527	0.8527	0.8683	0.8516	0.8939
CNN-SVM-Shi [41]	0.9111	0.9111	0.9142	0.9104	0.9030
The proposed CNN	0.9139	0.9139	0.9375	0.9152	0.9061
The proposed CNN-SVM	0.9611	0.9611	0.9686	0.9603	0.9576

**Table 5 foods-10-00795-t005:** Results of F1 scores for classifications by five models.

Class	SVM	CNN-Shi [41]	CNN-SVM-Shi [41]	The Proposed CNN	The Proposed CNN-SVM
HSMF	0.8302	0.9310	0.9474	0.9474	0.9777
YDMF	0.5000	0.8462	0.9667	0.9355	0.9731
EMMF	0.8000	0.9375	0.9677	1.0000	1.0000
LXMF	0.8000	1.0000	1.0000	1.0000	1.0000
QSMF	0.8000	0.6441	0.7234	0.6957	0.8924
MTMF	0.8000	0.8333	0.8364	0.8519	0.9108
XYMJ	0.7532	0.8150	0.9166	0.9474	0.9931
GLMJ	0.8000	0.7937	0.8824	0.8696	0.9375
DYMJ	0.8000	0.9231	1.0000	1.0000	1.0000
GZMJ	0.8000	0.8824	0.8955	0.9016	0.9438
QSMJ	0.5455	0.6129	0.7887	0.8333	0.8955
ZYMJ	0.8000	1.0000	1.0000	1.0000	1.0000

## Data Availability

The data presented in this study are available at Figshare (https://doi.org/10.6084/m9.figshare.14375927.v1 (accessed on 16 January 2021)).

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
