# Peer review of "A Machine Learning Method for the Fine-Grained Classification of Green Tea with Geographical Indication Using a MOS-Based Electronic Nose"

_foods, 2021, doi:10.3390/foods10040795_

Round 1

Reviewer 1 Report

The paper is interesting and includes a notable amount of work. My comments are mainly aimed at tightening up the logic and clarity of what you have communicated and tested. I summarize my comments below based on the order I found them in the paper.

Furthermore, the paper is well presented and written in good language and style. There are some minor revisions.

Some of them are listed below

Introduction:

Line 51-52: After the sentence "... used to analyze foods and beverages", please add a reference:

  1. Mariateresa Russo, Demetrio Serra, Francesca Suraci, Rosa Di Sanzo, Salvatore Fuda, Santo Postorino, “The potential of e-nose aroma profiling for identifying the geographical origin of licorice (glycyrrhiza glabra l.) Roots”, Food Chemistry, 165, 2014, 467–474
  2. Mariateresa Russo, Rosa Di Sanzo, Vittoria Cefaly, Sonia Carabetta, Demetrio Serra, Salvatore Fuda, “Non-destructive flavour evaluation of red onion (Allium cepa l.) Ecotypes: an electronic-nose-based approach” Food Chemistry, 2013, 141, pp. 896–899 ISSN: 0308-8146

Line 62-64: Please add some references

Line 96-98: The aim of this study is not clear, it should be more specific. Did you want only classify the twelve tea samples? Or, did you want to obtain a model to discriminate the tea made to a protected designation of origin areas in China from the other similar samples produced in other areas of China?

Materials and Methods:

Line 132 – 134: Are all samples from the same year of production?

Line 142 – 142: What temperature?

Proposed Method

Line 212: The sentence “ … a large amount of data is needed for CNN model training and may not be available in an E-nose study”. In my opinion, if any statistical model is applied, a significant amount of data is required.

Material and methods:

Section 4.1: did you try to see if the PCA was able to discriminate only the samples of protected designation of origin areas from the other samples?

Author Response

Original Manuscript ID: foods-1096164     

Original Article Title: “A machine learning method for the fine-grained classification of green tea with protected designation of origin using an electronic nose

Reviewer #1 (Comments and Suggestions for Authors):
The paper is interesting and includes a notable amount of work. My comments are mainly aimed at tightening up the logic and clarity of what you have communicated and tested. I summarize my comments below based on the order I found them in the paper. Furthermore, the paper is well presented and written in good language and style. There are some minor revisions. Some of them are listed below.

Introduction:

-Line 51-52: After the sentence "... used to analyze foods and beverages", please add a reference:

  1. Mariateresa Russo, Demetrio Serra, Francesca Suraci, Rosa Di Sanzo, Salvatore Fuda, Santo Postorino, “The potential of e-nose aroma profiling for identifying the geographical origin of licorice (glycyrrhiza glabra l.) Roots”, Food Chemistry, 165, 2014, 467–474
  2. Mariateresa Russo, Rosa Di Sanzo, Vittoria Cefaly, Sonia Carabetta, Demetrio Serra, Salvatore Fuda, “Non-destructive flavour evaluation of red onion (Allium cepa l.) Ecotypes: an electronic-nose-based approach” Food Chemistry, 2013, 141, pp. 896–899 ISSN: 0308-8146

-Line 62-64: Please add some references

-Line 96-98: The aim of this study is not clear, it should be more specific. Did you want only classify the twelve tea samples? Or, did you want to obtain a model to discriminate the tea made to a protected designation of origin areas in China from the other similar samples produced in other areas of China?

Materials and Methods:

-Line 132 – 134: Are all samples from the same year of production?

-Line 142 – 142: What temperature?

Proposed Method

-Line 212: The sentence “ … a large amount of data is needed for CNN model training and may not be available in an E-nose study”. In my opinion, if any statistical model is applied, a significant amount of data is required.

Material and methods:

-Section 4.1: did you try to see if the PCA was able to discriminate only the samples of protected designation of origin areas from the other samples?

Dear reviewer,

On the arrival of the Chinese Spring Festival, extend you all my best wishes for your perfect health and lasting prosperity.

Thanks for your supporting. Our answers for your questions are as follows:

Reviewer #1, Concern # 1:

- Line 51-52: After the sentence "... used to analyze foods and beverages", please add a reference:

  1. Mariateresa Russo, Demetrio Serra, Francesca Suraci, Rosa Di Sanzo, Salvatore Fuda, Santo Postorino, “The potential of e-nose aroma profiling for identifying the geographical origin of licorice (glycyrrhiza glabra l.) Roots”, Food Chemistry, 165, 2014, 467–474
  2. Mariateresa Russo, Rosa Di Sanzo, Vittoria Cefaly, Sonia Carabetta, Demetrio Serra, Salvatore Fuda, “Non-destructive flavour evaluation of red onion (Allium cepa l.) Ecotypes: an electronic-nose-based approach” Food Chemistry, 2013, 141, pp. 896–899 ISSN: 0308-8146

Author response: Dear reviewer, thanks a lot for your attentions and suggestions.

According to your suggestions, we have added the two references for this sentence in Section 1. Introduction (Line 50-52, Page 2, Clean revised version). Now the readability of our manuscript is increased. Thanks again.

“…As a powerful odor analysis device, the E-nose has been increasingly used to analyze foods and beverages [14-18]…”

References:

[17] Mariateresa Russo, Demetrio Serra, Francesca Suraci, Rosa Di Sanzo, Salvatore Fuda, Santo Postorino, “The potential of e-nose aroma profiling for identifying the geographical origin of licorice (glycyrrhiza glabra l.) Roots”, Food Chemistry, 165, 2014, 467–474

[18] Mariateresa Russo, Rosa Di Sanzo, Vittoria Cefaly, Sonia Carabetta, Demetrio Serra, Salvatore Fuda, “Non-destructive flavour evaluation of red onion (Allium cepa l.) Ecotypes: an electronic-nose-based approach” Food Chemistry, 2013, 141, pp. 896–899 ISSN: 0308-8146

Reviewer#1, Concern # 2:

- Line 62-64: Please add some references

Author response: Dear reviewer, thanks a lot for your attentions and suggestions.

According to your suggestions, we have added six references for the sentence in Section 1. Introduction (Line 59-62, Page 2, Clean revised version).

“…Although the E-nose has been successfully applied for analyzing teas, most studies primarily focused on distinguishing teas in different categories [19,22,23] or teas in the same category from different production areas or with different qualities [20,21,24] …”

Now the readability of our manuscript is increased. Thanks again.

References:

[19] Liu, H.; Li, Q.; Gu, Y. Convenient and accurate method for the identification of Chinese teas by an electronic nose. Qual. Assur. Saf. Crops Foods 2019, 11, 79-88.

[22] Gao, T.; Wang, Y.; Zhang, C.; Pittman, Z.A.; Oliveira, A.M.; Fu, K.; Willis, B.G. Classification of tea aromas using multi-nanoparticle based chemiresistor arrays. Sensors 2019, 19, 2547.

[23] Ruengdech, A.; Siripatrawan, U. Visualization of mulberry tea quality using an electronic sensor array, SPME-GC/MS, and sensory evaluation. Food Biosci. 2020, 36, 100593.

[20] Banerjee, M.B.; Roy, R.B.; Tudu, B.; Bandyopadhyay, R.; Bhattacharyya, N. Black tea classification employing feature fusion of E-Nose and E-Tongue responses. J. Food Eng. 2019, 244, 55-63.

[21] Lu, X.; Wang, J.; Lu, G.; Lin, B.; Chang, M.; He, W. Quality level identification of West Lake Longjing green tea using electronic nose. Sens. Actuators B Chem. 2019, 301, 127056.

[24] Wang, X.; Gu, Y.; Liu, H. A transfer learning method for the protection of geographical indication in China using an electronic nose for the identification of Xihu Longjing tea. IEEE Sens. J. 2021.

Reviewer#1, Concern # 3:

- Line 96-98: The aim of this study is not clear, it should be more specific. Did you want only classify the twelve tea samples? Or, did you want to obtain a model to discriminate the tea made to a protected designation of origin areas in China from the other similar samples produced in other areas of China?

Author response: Dear reviewer, thanks a lot for your attentions and suggestions.

  1. Thanks for your attentions to the aim of our study. We are so sorry that making your misunderstanding on the aim of our study due to our not rich English. After carefully thinking, we replaced the title of our manuscript from “A machine learning method for the fine-grained classification of green tea with protected designation of origin using an electronic nose” to “A machine learning method for the fine-grained classification of green tea with geographical indication using an MOS-based electronic nose”. We think the revised title is more suitable for showing the aim of our study. From the title, there are two aims as shown in follows:
  2. We proposed a machine learning framework (named CNN-SVM) which is consisted of a CNN backbone and a SVM classifier.
  3. We presented the fine-grained classification of green teas with geographical indication by means of the proposed CNN-SVM framework. The data of the teas’ samples were collected from a commercial MOS-based E-nose (called PEN3).
  4. In this study, we wanted to propose a machine learning model for the classification of the GTSGI and the identification of the FGTSGI. The descriptions of the GTSGI and FGTSGI are shown as follows:

Chinese green tea, which has rich categories, is known for its health functional properties. Each green tea category includes some sub-categories with geographical indications (GTSGI). Several GTSGI with high quality planted in specific areas, labeled famous GTSGI (FGTSGI), are expensive. In our study, the proposed CNN-SVM model was verified in the twelve green tea sub-categories (belong to two main green tea categories). Our proposed CNN-SVM model presents good performances and strong robustness for the fine-grained classification of the twelve GTSGI. Furthermore, extra experiments, identifying HSMF (the most famous and expensive Maofeng green tea sub-category) and XYMJ (the most famous and expensive Maojian green tea sub-category) were performed. The identification results for the two FGTSGI show the CNN-SVM achieved the highest F1 scores (97.77% for HSMF and 99.31% for XYMJ) among the five machine learning models.

  1. As far as we known, there are many studies that primarily focused on distinguishing teas in different categories [1-3] and teas in the same category from different production areas or with different qualities [4-6], respectively. Few studies have focused on the fine-grained classification of tea in different categories (categories) and from different production areas (sub-categories) simultaneously. Green teas usually have light aromas and these aromas have small differences among different green tea categories. An increasing number of sub-categories has complicated the classification of these similar samples, because there are not only tiny differences in green teas among categories but also much tinier differences among sub-categories. In our study, we used two main green teas’ categories (Maofeng tea category and Maojian tea category). Each category contained six sub-categories (six Maofeng tea sub-categories and six Maojian tea sub-categories), respectively. All the tea samples were picked 14 days before the Pure Brightness Festival (around April 5 or 6) and were called “before-brightness tea”. As we introduced in our manuscript, the twelve green teas sub-categories, belong to two categories, not only have a high similarity in the aroma among the sub-categories, but also have tiny differences between the two categories. We proposed a machine learning method, named CNN-SVM, for the fine-grained classification of green tea with geographical indication using an electronic nose. In our work, we analyzed the network structure of the proposed CNN-SVM and verified the effectiveness of the CNN-SVM via the twelve green teas. Furthermore, we presented the good performance of our proposed CNN-SVM for the identification of the two most famous and expensive teas with protected designation of origin (HSMF and XYMJ) via extra experiments. The proposed framework had the best performance for both the classification of the GTSGI and the identification of the FGTSGI.
  2. The effectiveness of our proposed CNN-SVM model was demonstrated on the classification of the twelve green teas with high accuracy and strong robustness. As we introduced before, green teas usually have light aromas which are much weaker than those of the other common tea categories in China, such as oolong tea, black tea, and jasmine tea. The study provides a novel insight for distinguishing multiple highly similar green tea and might have a potential for other tea’s fine-grained classification tasks in the future.
  3. According to your suggestions, we have made some changes in our manuscript and the details are shown as follows:
  4. We have revised the title of our manuscript with “A machine learning method for the fine-grained classification of green tea with geographical indication using an MOS-based electronic nose”.
  5. We rewritten the Abstract of our manuscript to present the aims and contributions of our study more clearly. The revised version is shown in Section Abstract (Line 1-15, Page 1, Clean revised version).

“Chinese green tea, which has rich categories, is known for its health functional properties. Each green tea category includes some sub-categories with geographical indications (GTSGI). Several GTSGI with high quality planted in specific areas, labeled famous GTSGI (FGTSGI), are expensive. However, the fine-grained classification of the GTSGI is complex because of their subtle differences. This study proposes a novel framework consisting of a convolutional neural network backbone (CNN backbone) and a support vector machine classifier (SVM classifier), namely, CNN-SVM for the classification of Maofeng green tea categories (six sub-categories) and Maojian green tea categories (six sub-categories) using electronic nose data. A multi-channel input matrix was constructed for the CNN backbone to extract deep features from different sensor signals. An SVM classifier was employed to improve the classification performance due to its high discrimination ability for small sample sizes. The effectiveness of this framework was verified by comparing it with other four machine learning models (SVM, CNN-Shi, CNN-SVM-Shi, and CNN). The proposed framework had the best performance for both the classification of the GTSGI and the identification of the FGTSGI. The high accuracy and strong robustness of the CNN-SVM show its potential for the fine-grained classification of multiple highly similar teas.”

  1. We almost completely rewritten the description of the Introduction to show the aim of our study more clearly.

c1. We have reconstructed the structure of our Introduction Section to improve the manuscript’s logic and readability.

c2. We have added more references on the description of the classification of green tea in Section 1. Introduction (Line 59-67, Page 2, Clean revised version).

…Although the E-nose has been successfully applied for analyzing teas, most studies primarily focused on distinguishing teas in different categories [19,22,23] or teas in the same category from different production areas or with different qualities [20,21,24]. Few studies focused on the fine-grained classification of tea in different categories (categories) and from different production areas (sub-categories) simultaneously. Green teas typically have light aromas with subtle differences between different green tea categories or sub-categories. These slight differences coupled with an increasing number of sub-categories have complicated the classification…

c3. We have added the main contributions of the proposed CNN-SVM framework in Section 1. Introduction (Line 96-111, Page 3, Clean revised version).

…This paper presents an effective approach for the fine-grained classification of green tea with geographical indication based on a novel deep learning framework using a MOS-based E-nose. The framework consists of a CNN backbone and a support vector machine (SVM) classifier and is referred to as CNN-SVM. This study makes the following contributions:

  1. A preprocessing scheme inspired by the multi-channel input to the CNN model used in image processing is proposed for the 10-channel E-nose’s data. The channel’s sensor data are converted into a single matrix, and the combined matrixes of the 10 channels represent the multi-channel data.
  2. The network structure of the proposed CNN-SVM, a CNN backbone with a SVM classifier, is analyzed. In the framework, the deep CNN is designed to mine the aroma features of the tea, and the SVM is employed to classify the data (small sample sizes) to improve the classification performance.
  3. A comprehensive study on the fine-grained classification of green tea with geographical indication is presented, demonstrating the high accuracy and strong robustness of the proposed CNN-SVM framework.”
  4. We have made extra experiments and added a Table 5 to present the good performance of our proposed CNN-SVM for the identification of the two FGTSGI (HSMF and XYMJ, which are the two most famous and expensive teas with protected designation of origin) in Section 4.2. Comparison of the Classification Results of five Models (Line 385-392, Page 14, Clean revised version).

…In order to obtain a more practical evaluation of the CNN-SVM framework while further proving its usability, an experiment was performed (identification of the two FGTSGI from the 12 GTSGI). The F1-scores of the classes are listed in Table 5. The proposed CNN-SVM model outperformed the other four models for the identification of the two most famous and expensive teas with protected designation of origin (F1 scores of 97.77% for HSMF and 99.31% for XYMJ). The experimental results indicated that the model could accurately distinguish HSMF and XYMJ from the other Maofeng and Maojian tea sub-categories.

  1. We updated the Conclusion Section of our manuscript to summarize the novelty of our study more clearly. The revised version is shown in Section 5. Conclusion (Line 420-433, Page 15, Clean revised version).

“4. Comparing with other four machine learning models (SVM, CNN-Shi, CNN-SVM-Shi, and CNN), good performances of the proposed CNN-SVM were obtained for the classification of the GTSGI, with the best scores of the five evaluation metrics: accuracy of 96.11%, recall of 96.11%, precision of 96.86%, F1 score of 96.03%, and Kappa score of 95.76%, respectively. Excellent performances were acquired for the identification of the FGTSGI, with the highest score of F1 scores of 97.77% (for HSMF) and 99.31% (for XYMJ), respectively. These experimental results demonstrated the effectiveness of the CNN-SVM both for the classification of the GTSGI and the identification of the FGTSGI.

  1. This study provides novel insights into fine-grained classification using an E-nose system, protecting consumers’ interests and producers’ reputations. The proposed method also shows a strong potential for distinguishing multiple highly similar green tea and may be applicable to the fine-grained classifications of other teas in the future.”

In summary, our proposed CNN-SVM model presents the good performance not only for the classification of GTSGI, but also for the identification of FGTSGI.

We have improved the grammar and language by a 3rd party service for language polishing. Now, a deep revision of the manuscript is uploaded for your review.

According to your suggestions, now the aims and novelty of our study are presented more clearly, and the readability of our manuscript is strongly increased. We really hope that we could have a deeper communication on science with you in the future. Thanks again for the very professional advices.

References:

  • Liu, H.; Li, Q.; Gu, Y. Convenient and accurate method for the identification of Chinese teas by an electronic nose. Qual. Assur. Saf. Crops Foods 2019, 11, 79-88.
  • Gao, T.; Wang, Y.; Zhang, C.; Pittman, Z.A.; Oliveira, A.M.; Fu, K.; Willis, B.G. Classification of tea aromas using multi-nanoparticle based chemiresistor arrays. Sensors 2019, 19, 2547.
  • Ruengdech, A.; Siripatrawan, U. Visualization of mulberry tea quality using an electronic sensor array, SPME-GC/MS, and sensory evaluation. Food Biosci. 2020, 36, 100593.
  • Banerjee, M.B.; Roy, R.B.; Tudu, B.; Bandyopadhyay, R.; Bhattacharyya, N. Black tea classification employing feature fusion of E-Nose and E-Tongue responses. J. Food Eng. 2019, 244, 55-63.
  • Lu, X.; Wang, J.; Lu, G.; Lin, B.; Chang, M.; He, W. Quality level identification of West Lake Longjing green tea using electronic nose. Sens. Actuators B Chem. 2019, 301, 127056.
  • Wang, X.; Gu, Y.; Liu, H. A transfer learning method for the protection of geographical indication in China using an electronic nose for the identification of Xihu Longjing tea. IEEE Sens. J. 2021.

Reviewer#1, Concern # 4:

- Line 132 – 134: Are all samples from the same year of production?

Author response: Dear reviewer, thanks a lot for your attentions. We are so sorry for the lack of the important details.

  1. In this study, all the tea samples were picked and processed 14 days before the Pure Brightness Festival (around April 5 or 6) in 2019. We have revised the description of tea samples in Section 2.2 Experimental Samples and Conditions (Line 138-142, Page 4, Clean revised version).

“…In this study, all the tea samples (twelve teas with special-class levels) were purchased from specialty stores approved by the China Tea Marketing Association. The tea was picked and processed 14 days before the Pure Brightness Festival (around April 5 or 6) in 2019 and is called “before-brightness tea”. The details of the tea samples are listed in Table 1. Photos of the 12 green teas are shown in Fig. 3...”

  1. We are so sorry that we had some mistakes about the teas’ prices in Table 1. We lost the decimal points for the price values of teas by our carelessness. We have revised the teas’ prices in Table 1 (Table 1, Page 5, Clean revised version).

Now the rigor of our manuscript is increased according to your suggestions. We hope the revised version can meet your requirements.

Reviewer#1, Concern # 5:

- Line 142 – 142: What temperature?

Author response: Dear reviewer, thanks a lot for your attentions.

As we introduced in Section 2.2 Experimental Samples and Conditions (Line 146-147, Page 4, Clean revised version), all the experiments were conducted in the authors’ laboratory at a temperature of 25 °C ± 1 °C and a humidity level of 40% ± 2%.

“…The experiments were conducted for 12 days in the authors’ laboratory at a temperature of 25 °C ± 1 °C and a humidity level of 40% ± 2%...”

Reviewer#1, Concern # 6:

- Line 212: The sentence “ … a large amount of data is needed for CNN model training and may not be available in an E-nose study”. In my opinion, if any statistical model is applied, a significant amount of data is required.

Author response: Dear reviewer, thanks a lot for your attentions.

We are so sorry that making your misunderstanding because of the ambiguity in English. But we would like to make some explanations on your question.

  1. As far as we known, there are two main types of statistical models, including deep learning models and traditional statistical models [1]. As you said, the statistical models require a significant amount of data. But the deep learning models needs a much larger amount of data compared with the traditional statistical models.
  2. A CNN is a deep learning model designed to process data in multiple arrays by extracting local and global features [2]. Although CNN has many advantages, it requires a large amount of training data [3]. Many researchers have applied the CNN in the field of image processing based on a large number of images. For example, Liang et al. [4] used the MURA dataset (including 40,895 multi-view radiographic images) to train the multi-scale CNN model for the detection of musculoskeletal abnormalities. Xie et al. [5] verified the effectiveness of the ResNext model on the ImageNet-5K dataset (including 6.8 million images) and the COCO dataset (including 118,287 images). A large amount of data is needed for CNN model training because of the deep network structure.
  3. In this study, we hope to propose a CNN network to classify green teas by means of E-nose data. When the data is not sufficient, the CNN model cannot be trained completely and could cause the problem of overfitting. The trained model may tend to overfit on the training data and perform not good enough on the testing data. In order to solve the problem, a SVM classifier was employed by us. In the practical applications of the E-nose, a large sample data acquisition may not be easy. But we hope our proposed model can achieve a good performance for small sample sizes based on the E-nose. As we introduced in the manuscript, according to the principle of structural risk minimization, the SVM has good pattern recognition ability for small sample sizes of data. Thus, a CNN-SVM framework consisting of a CNN backbone and a SVM classifier was proposed in this study. In the framework, a deep CNN is designed to mine the olfactory features of the E-nose signals, and the SVM is used to improve the classification performance. The effectiveness of this framework was verified by comparing with four machine learning models (SVM, CNN-Shi, CNN-SVM-Shi, and CNN) with the best performance on all five evaluation metrics (accuracy, recall, precision, F1-score, and Kappa score). The results show that the proposed model has strong potential for the fine-grained classification of multiple highly similar teas.
  4. According to your suggestions, we have revised the description in Section 3. Proposed Method (Line 219-228, Page 8, Clean revised version).

“…A CNN with an appropriate structure can mine the deep features of the input data and has a flexible form. However, more data are required for CNN model training than for a traditional statistical model [40], and the data volume may not be sufficient in the practical applications of E-nose, potentially resulting in overfitting. The SVM is considered a good classifier for small sample sizes, but it is difficult to select or extract valid features using the SVM. In this study, we combined the advantages of these two algorithms and proposed the CNN-SVM framework. The deep CNN structure was used to mine the aroma features of the tea from the raw data of the E-nose, and the SVM was used to classify the data to achieve high classification accuracy. The proposed CNN-SVM framework consisting of a CNN backbone and an SVM classifier is shown in Fig. 7…”

We hope our responses can remove your concerns and the revised version can meet your requirements.

References:

  • Munir, M.; Siddiqui, S.A.; Chattha, M.A.; Dengel, A.; Ahmed, S. FuseAD: unsupervised anomaly detection in streaming sensors data by fusing statistical and deep learning models. Sensors 2019, 19, 2451.
  • Lecun Y.; Bengio Y.; Hinton G. Deep learning. Nature 2015, 521, 436.
  • Shi, Y.; Gong, F.; Wang, M.; Liu, J.; Wu, Y.; Men, H. A deep feature mining method of electronic nose sensor data for identifying beer olfactory information. J. Food Eng. 2019, 263, 437-445.
  • Liang, S.; Gu, Y. Towards Robust and Accurate Detection of Abnormalities in Musculoskeletal Radiographs with a Multi-Network Model. Sensors 2020, 20, 3153.
  • Xie, S.; Girshick, R.; Dollár, P.; Tu, Z.; He, K. Aggregated residual transformations for deep neural networks. In Proceedings of the IEEE conference on computer vision and pattern recognition (CVPR), Honolulu, Hawaii, USA, 21-26 July, 2017; pp. 1492-1500.

Reviewer#1, Concern # 7:

- Section 4.1: did you try to see if the PCA was able to discriminate only the samples of protected designation of origin areas from the other samples?

Author response: Dear reviewer, thanks a lot for your attentions.

  1. We had tried to use PCA to illustrate the data distribution of the two FGTSGI tea samples (Xinyang Maojian tea and Huangshan Maofeng tea) and other GTSGI tea samples, as is shown in Fig 1 below. The data distribution of the tea samples for this situation is same with that in the original PCA score plot (Figure 10, Page 12, Clean revised version), where all the samples overlapped heavily. It is still difficult to distinguish them visually in the two-dimensional projection space.

Fig1. Projection of the first two principal components of the PCA of the tea samples (HSMF, XYMJ, and other tea samples) (Please see in the uploaded PDF)

  1. The poor PCA results on distinguishments indicated that the aroma features of the teas were complex and the differences among the tea varieties were relatively tiny. Some meaningful feature information for the fine-grained classification was lost. Therefore, we proposed the CNN-SVM framework to mine the deep aroma features of the teas for the fine-grained classification of the twelve green teas.
  2. We have added a Table 5 to present the good performance of our proposed CNN-SVM for the identification of the two FGTSGI teas in Section 4.2. Comparison of the Classification Results of five Models (Line 385-392, Page 14, Clean revised version).

…In order to obtain a more practical evaluation of the CNN-SVM framework while further proving its usability, an experiment was performed (identification of the two FGTSGI from the 12 GTSGI). The F1-scores of the classes are listed in Table 5. The proposed CNN-SVM model outperformed the other four models for the identification of the two most famous and expensive teas with protected designation of origin (F1 scores of 97.77% for HSMF and 99.31% for XYMJ). The experimental results indicated that the model could accurately distinguish HSMF and XYMJ from the other Maofeng and Maojian tea sub-categories.

We hope our responses can remove your concerns and the revised version can meet your requirements.

Reviewer 2 Report

Most of the Figures are theoretical. The experimental design is weak and not informative. This paper needs further data and work.

The introduction is very long and not focused. The methodology is nice but there no results. The authors gave almost 2 pages of results and discussion from 16 pages.

Author Response

Original Manuscript ID: foods-1096164     

Original Article Title: “A machine learning method for the fine-grained classification of green tea with protected designation of origin using an electronic nose

Reviewer #2 (Comments for Authors):
This paper has no novelty. Most of the Figures are theoretical and not brining anything to science. The experimental design is weak and not informative. This paper needs further data and work.

The introduction is very long and not focused. The methodology is nice but there no results. The authors gave almost 2 pages of results and discussion from 16 pages.

Dear reviewer,

On the arrival of the Chinese Spring Festival, extend you all my best wishes for your perfect health and lasting prosperity.

Thanks for your supporting. Our answers for your questions are as follows:

Author response: Dear reviewer, thanks for your attentions.

We are so sorry that making your misunderstanding by our not rich English. After carefully thinking, we revised our manuscript to increase the its readability.

1. We replaced the title of our manuscript from “A machine learning method for the fine-grained classification of green tea with protected designation of origin using an electronic nose” to “A machine learning method for the fine-grained classification of green tea with geographical indication using an MOS-based electronic nose”. We think the revised title is more suitable for showing the aim of our study.

2. We rewritten the Abstract of our manuscript to present the aims and contributions of our study more clearly. The revised version is shown in Section Abstract (Line 1-15, Page 1, Clean revised version).

“Chinese green tea, which has rich categories, is known for its health functional properties. Each green tea category includes some sub-categories with geographical indications (GTSGI). Several GTSGI with high quality planted in specific areas, labeled famous GTSGI (FGTSGI), are expensive. However, the fine-grained classification of the GTSGI is complex because of their subtle differences. This study proposes a novel framework consisting of a convolutional neural network backbone (CNN backbone) and a support vector machine classifier (SVM classifier), namely, CNN-SVM for the classification of Maofeng green tea categories (six sub-categories) and Maojian green tea categories (six sub-categories) using electronic nose data. A multi-channel input matrix was constructed for the CNN backbone to extract deep features from different sensor signals. An SVM classifier was employed to improve the classification performance due to its high discrimination ability for small sample sizes. The effectiveness of this framework was verified by comparing it with other four machine learning models (SVM, CNN-Shi, CNN-SVM-Shi, and CNN). The proposed framework had the best performance for both the classification of the GTSGI and the identification of the FGTSGI. The high accuracy and strong robustness of the CNN-SVM show its potential for the fine-grained classification of multiple highly similar teas.”

3. We almost completely rewritten the description of the Introduction to show the aim of our study more clearly. The revised version is shown as follows:

3.1 We have reconstructed the structure of our Introduction Section to improve the manuscript’s logic and readability.

3.2 We have added more references on the description of the classification of green tea in Section 1. Introduction (Line 59-67, Page 2, Clean revised version).

“…Although the E-nose has been successfully applied for analyzing teas, most studies primarily focused on distinguishing teas in different categories [19,22,23] or teas in the same category from different production areas or with different qualities [20,21,24]. Few studies focused on the fine-grained classification of tea in different categories (categories) and from different production areas (sub-categories) simultaneously. Green teas typically have light aromas with subtle differences among different green tea categories or sub-categories. These slight differences coupled with an increasing number of sub-categories have complicated the classification…”

3.3 We have added the main contributions of the proposed CNN-SVM framework in Section 1. Introduction (Line 96-111, Page 3, Clean revised version).

“…This paper presents an effective approach for the fine-grained classification of green tea with geographical indication based on a novel deep learning framework using a MOS-based E-nose. The framework consists of a CNN backbone and a support vector machine (SVM) classifier and is referred to as CNN-SVM. This study makes the following contributions:

  1. A preprocessing scheme inspired by the multi-channel input to the CNN model used in image processing is proposed for the 10-channel E-nose’s data. The channel’s sensor data are converted into a single matrix, and the combined matrixes of the 10 channels represent the multi-channel data.
  2. The network structure of the proposed CNN-SVM, a CNN backbone with a SVM classifier, is analyzed. In the framework, the deep CNN is designed to mine the aroma features of the tea, and the SVM is employed to classify the data (small sample sizes) to improve the classification performance.
  3. A comprehensive study on the fine-grained classification of green tea with geographical indication is presented, demonstrating the high accuracy and strong robustness of the proposed CNN-SVM framework.”

4. We updated the Conclusion section of our manuscript to summarize the novelty of our study more clearly. The revised version is shown in Section 5. Conclusion (Line 420-433, Page 15, Clean revised version).

“4. Comparing with other four machine learning models (SVM, CNN-Shi, CNN-SVM-Shi, and CNN), good performances of the proposed CNN-SVM were obtained for the classification of the GTSGI, with the best scores of the five evaluation metrics: accuracy of 96.11%, recall of 96.11%, precision of 96.86%, F1 score of 96.03%, and Kappa score of 95.76%, respectively. Excellent performances were acquired for the identification of the FGTSGI, with the highest score of F1 scores of 97.77% (for HSMF) and 99.31% (for XYMJ), respectively. These experimental results demonstrated the effectiveness of the CNN-SVM both for the classification of the GTSGI and the identification of the FGTSGI.

5. This study provides novel insights into fine-grained classification using an E-nose system, protecting consumers’ interests and producers’ reputations. The proposed method also shows a strong potential for distinguishing multiple highly similar green tea and may be applicable to the fine-grained classifications of other teas in the future.”

5. Anyway, we would like to make some explanations to you for the novelty of our study. The explanations are shown as follows:

1. By the revised title of our manuscript (A machine learning method for the fine-grained classification of green tea with geographical indication using an MOS-based electronic nose), two aims of our study are presented: 1. We proposed a machine learning framework (named CNN-SVM) which is consisted of a CNN backbone and a SVM classifier. 2. We presented the fine-grained classification of green tea with geographical indication by means of the proposed CNN-SVM framework. The data of the teas’ samples were collected from a commercial MOS-based E-nose.

2. In this study, we wanted to propose a machine learning model for the classification of the GTSGI and the identification of the FGTSGI. The descriptions of the GTSGI and FGTSGI are shown as follows:

Chinese green tea, which has rich categories, is known for its health functional properties. Each green tea category includes some sub-categories with geographical indications (GTSGI). Several GTSGI with high quality planted in specific areas, labeled famous GTSGI (FGTSGI), are expensive. In our study, the proposed CNN-SVM model was verified in the twelve green tea sub-categories (belong to two main green tea categories). Our proposed CNN-SVM model presents good performances and strong robustness for the fine-grained classification of the twelve GTSGI. Furthermore, extra experiments, identifying HSMF (the most famous and expensive Maofeng green tea sub-category) and XYMJ (the most famous and expensive Maojian green tea sub-category) were performed. The identification results for the two FGTSGI show the CNN-SVM achieved the highest F1 scores (97.77% for HSMF and 99.31% for XYMJ) among the five machine learning models.

3. As we introduced in our manuscript, although the E-nose has been successfully applied for analyzing teas, most studies primarily focused on distinguishing teas in different categories or teas in the same category from different production areas or with different qualities. Few studies focused on the fine-grained classification of teas in different categories (categories) and from different production areas (sub-categories) simultaneously. As far as we known, green teas usually have light aromas and these aromas have tiny differences among different green tea categories or sub-categories. An increasing number of sub-categories has complicated the classification of these similar samples, because there are not only tiny differences in green teas among categories but also much tinier differences among sub-categories. In this study, we proposed a CNN-SVM framework consisting of the CNN backbone and the SVM classifier to perform the classification using the 10-channel MOS-based E-nose data. The novel structure of the CNN backbone based on ResNeXt was effective for extracting the deep features from the different channels automatically. The SVM classifier improved the generalization ability of the CNN model and increased the classification accuracy from 91.39% (CNN backbone + CNN classifier) to 96.11% (CNN backbone + SVM classifier) among the twelve green teas due to its good discrimination ability for small sample sizes. In order to obtain a more practical evaluation of the CNN-SVM framework while further proving its usability, an extra experiment was performed (identification of the two FGTSGI from the 12 GTSGI). The proposed CNN-SVM model outperformed the other four models for the identification of the two most famous and expensive teas with protected designation of origin (F1 scores of 97.77% for HSMF and 99.31% for XYMJ). The experimental results indicated that the model could accurately distinguish HSMF and XYMJ from the other Maofeng and Maojian tea sub-categories.

4. As far as we known, CNNs are usually used in the field of image processing. Inspired by the multi-channel input to the CNN model, a preprocessing scheme of the 10-channel E-nose’s data was proposed. The proposed data preprocessing scheme not only considers the data structure of the E-nose data, but also considers the data correlation among the 10-channel sensors (the high cross-sensitivity of the sensor array in the E-nose). Each channel sensor’s data was converted into a single matrix, and the combined matrixes of 10 channels represented the multi-channel data.

5. The effectiveness of our proposed CNN-SVM model was demonstrated on the classification of the twelve green teas with high accuracy and strong robustness. As we introduced before, green teas usually have light aromas which are much weaker than those of the other common tea categories in China, such as oolong tea, black tea, and jasmine tea. The study provides a novel insight for distinguishing multiple highly similar green tea and might have a potential for other tea’s fine-grained classification tasks in the future.

6. In summary, our proposed CNN-SVM model presents the good performance not only for the classification of GTSGI, but also for the identification of FGTSGI.

7. We have improved the grammar and language by a 3rd party service for language polishing. Now, a deep revision of the manuscript is uploaded for your review.

According to your suggestions, now the aims and novelty of our study are presented more clearly, and the readability of our manuscript is strongly increased. We really hope that we could have a deeper communication on science with you in the future. Thanks again for your supporting to our work.

Reviewer 3 Report

The manuscript "A machine learning method for the fine-grained classification of green tea with protected designation of origin using an electronic nose " presents the classification of two varieties of tea (Maofeng tea and Maojian tea) using MOS-array e-nose and CNN-SVM classifier.

In the text I found a few elements that should be clarified and improved, namely:

1. In section 2.2. authors use the term "e-nose calibration" to describe sensors regeneration. The calibration is completely different.

2. What was the signal collection system from sensors? Was it based on a voltage divider? How was the conductivity measured? When analyzing Figure 5, I am not sure if the graph shows the change in conductivity or resistance.

3. The whole method of data analysis is described in detail, which is undoubtedly the strong point of the manuscript.

There are also some editorial notes:

Line 118, 124: standardize the notation of geographic coordinates

Line 144/145: "clean the surface" or "calibration" - better use "for sensor regeneration"

Author Response

Original Manuscript ID: foods-1096164     

Original Article Title: “A machine learning method for the fine-grained classification of green tea with protected designation of origin using an electronic nose

Reviewer #3 (Comments and Suggestions for Authors):
The manuscript "A machine learning method for the fine-grained classification of green tea with protected designation of origin using an electronic nose " presents the classification of two varieties of tea (Maofeng tea and Maojian tea) using MOS-array e-nose and CNN-SVM classifier.

In the text I found a few elements that should be clarified and improved, namely:

  1. In section 2.2. authors use the term "e-nose calibration" to describe sensors regeneration. The calibration is completely different.
  2. What was the signal collection system from sensors? Was it based on a voltage divider? How was the conductivity measured? When analyzing Figure 5, I am not sure if the graph shows the change in conductivity or resistance.
  3. The whole method of data analysis is described in detail, which is undoubtedly the strong point of the manuscript.

There are also some editorial notes:

Line 118, 124: standardize the notation of geographic coordinates

Line 144/145: "clean the surface" or "calibration" - better use "for sensor regeneration"

Considering the above, in my opinion, the article can be forwarded to the next stages of evaluation after minor revision.

Dear reviewer,

On the arrival of the Chinese Spring Festival, extend you all my best wishes for your perfect health and lasting prosperity.

Thanks for your supporting. Our answers for your questions are as follows:

Reviewer #3, Concern # 1 and Concern # 2:

- In section 2.2. authors use the term "e-nose calibration" to describe sensors regeneration. The calibration is completely different

- Line 144/145: "clean the surface" or "calibration" - better use "for sensor regeneration"

Author response: Dear reviewer, thanks for your attentions and suggestions. Please allow me answer the “Concern # 1 and Concern # 2” both here.

Thanks a lot for pointing out a better word here. We have replaced the “clean the surface” and the “calibration” with the “for sensor regeneration” in Section 2.2 Experimental Samples and Conditions (Line 150-166, Page 4, Clean revised version).

“…Before the measurement, valve 1 was closed, and valve 2 was opened for sensor regeneration. Clean air was pumped through filter 2 into the E-nose with a flow rate of 10 ml/s in the direction of In 2 for 100 s. After the sensor regeneration, valve 2 was closed, and valve 1 was opened for data collection. The tea sample's volatile gas in the sampler was pumped into the E-nose with a flow rate of 10 ml/s in the direction of In 1 to contact the sensor array for 100s. The gas molecules were adsorbed on the sensors' surface, changing the sensors' conductivity due to the redox reaction between the sensors and the gas. The sensors' conductivity eventually stabilized at a constant value when the adsorption was saturated. The collection stage lasted 100 s, and sampling continued at one sample per second. Figure 5 shows examples of the 10 sensors' response curves in the collection stage for the 12 tea samples. The response value (R) of the sensor was calculated with the equation R = G0/G, where G0 is the conductivity of the sensor in reference air (in the sensor regeneration step), and G is the conductivity of the sensor exposed to the sample vapor (in the data collection step). The sensor regeneration and data collection steps were repeated to obtain the raw data of 12 tea samples. Therefore, the dataset was obtained for 1440 measurements (12 tea samples 10 measuring times 12 days)…”

The readability of our revised manuscript is increased by means of your suggestions. Thanks so much. And we hope the revision could meet your requirements.

Reviewer#3, Concern # 3:

- What was the signal collection system from sensors? Was it based on a voltage divider? How was the conductivity measured? When analyzing Figure 5, I am not sure if the graph shows the change in conductivity or resistance.

Author response: Dear reviewer, thanks for your attentions.

1. After carefully thinking, we replaced the title of our manuscript from “A machine learning method for the fine-grained classification of green tea with protected designation of origin using an electronic nose” to “A machine learning method for the fine-grained classification of green tea with geographical indication using a MOS-based electronic nose”. We think the revised title is more suitable for showing the aim of our study. From the title, there are three aims as shown in follows:

a. We proposed a machine learning framework (named CNN-SVM) which is consisted of a CNN backbone and a SVM classifier.

b. We presented the fine-grained classification of green teas with geographical indication by means of the proposed CNN-SVM framework.

c. The data of the teas’ samples were collected from a commercial MOS-based E-nose (called PEN3).

2. The PEN3 used in our study is a commercial instrument (produced by “Airsense Analytics GmbH, Germany”) with 10-channel MOS-based sensors. As we introduced above, the main aim of our study is proposing a machine learning method for the fine-grained classification of green teas based on the commercial E-nose. So, we did not describe more details (details of working principle) about the PEN3 (the well-known commercial MOS-based E-nose) in our manuscript. But in order to remove your concerns, we would like to make some explanations as follows:

According to the handbook of the PEN3 (more details can be found on the website of the PEN3, www.airsense.com), the PEN3 is based on a metal-oxide gas sensor array which consisted of 10 MOS-based sensors. The working principle is shown as follows:

a. As you pointed out, the signal collection system from sensors is based on a voltage divider. For each MOS-based sensor, as shown in Fig 1, the system requires two input voltage, including the voltage supplied to the heater (VH) and the voltage supplied to the circuit (VC). When the heating resistance (RH) is supplied by the VH, the sensing resistance (RS) is heated to the optimum working temperature to detect the object gas. A load resistance (RL) is used to obtained the output voltage (VOUT).

Fig 1. Signal collection system. (Please see in the PDF)

Fig 2. An example of the sensing resistance’s sensitivity curve. (Please see in the PDF)

b. Fig 2 shows an example of the sensing resistance’s sensitivity curve, where the RS represents the sensor’s resistance value in object gas and the Ro represents the sensor’s resistance value in clean air. The interaction between sensor and gas will produce a redox reaction, which changes the resistance of sensor active materials, then changes the voltage of the voltage divider. Thus, the sensor’s conductivity in the clean air can be calculated by the formula G0 = 1/ Ro, and the sensor’s conductivity in the object gas can be calculated by the formula G = 1/ RS.

3. Figure 5 showed examples of the 10 sensors’ response curves in the data collection stage for the 12 tea samples. As the ordinate shown in Figure 5 (Figure 5, Page 6, Clean revised version), the graph shows the change in the ratio of the sensor’s conductivity and can be calculated by the formula R = G0/G, where R is the response value of the sensor, G0 is the conductivity of the sensor in reference air, and G is the conductivity of the sensor when exposed to the sample vapor.

4. We have revied the description of Figure 5 in Section 2.2 Experimental Samples and Conditions (Line 159-165, Page 4, Clean revised version).

“…Figure 5 shows examples of the 10 sensors' response curves in the collection stage for the 12 tea samples. The response value (R) of the sensor was calculated with the equation R = G0/G, where G0 is the conductivity of the sensor in reference air (in the sensor regeneration step), and G is the conductivity of the sensor exposed to the sample vapor (in the data collection step). The sensor regeneration and data collection steps were repeated to obtain the raw data of 12 tea samples…”

We hope our response can remove your concerns and the revision could meet your requirements.

Reviewer#3, Concern # 4:

- The whole method of data analysis is described in detail, which is undoubtedly the strong point of the manuscript.

Author response: Dear reviewer, thanks for your attentions and suggestions.

In this study, our main aim is proposing a machine learning method for the fine-grained classification of green teas by means of the MOS-based E-nose data. As far as we known, there are two principal approaches of building machine learning models: the model-driven (parametric) approach and the data-driven (non-parametric) approach [1]. The model-driven approach needs to design selectors (strategy or model to select samples from candidate data set) with parameter by handcraft feature or metric. For the data-driven approach, their selectors adopted deep architecture whose features are automatically generated but not by handcraft [2].

In our study, a CNN-SVM framework consisted of a CNN backbone and a SVM classifier, a data-driven model, was proposed for the fine-grained classification of the twelve green teas by means of a MOS-based E-nose. A preprocessing scheme inspired by the multi-channel input to the CNN model used in image processing was proposed for the 10-channel E-nose’s data. The channel’s sensor data are converted into a single matrix, and the combined matrixes of the 10 channels represent the multi-channel data. A deep CNN is designed to automatically mine the aroma features of the teas from the multi-channel data. The CNN can mine the potential information by automatically extracting local and global features and feed the information to the classifier providing a good performance. So, for the CNN-SVM model (a data-driven model), the preprocessing scheme of the input data is important.

According to your suggestions, we have adjusted the structure of Section 3 Proposed Method and added the Section 3.1 Method of Data Preprocessing to show our proposed preprocessing scheme in detail. (Line229-240, Page 9, Clean revised version). We have added the text “Data Preprocessing” in Figure 7 (Figure7, Page 9, Clean revised version) to highlight the data preprocessing step in our proposed methods. Since our manuscript is a latex format, the change traces of the revised figures cannot be directly displayed in the revised version. We have added the annotation in Figure 7 (Figure7, Page 9, Track Changes version). Thanks for your understanding.

“…The 10-channel sensor array in the PEN3 system has high cross-sensitivity because each sensor is sensitive to a specific substance. The E-nose data need to be preprocessed to consider the correlation between the sensors for the input of the CNN model. Therefore, data preprocessing is crucial to improve the classification accuracy. Inspired by the multi-channel input to the CNN model used in image processing, the raw data of the 10 sensors were converted into a 10-channel input for the CNN. As is shown in Fig.7, the matrix form of the original data was 10010, where 100 represents the number of sampling times in the data collection stage (100 s) for each sensor, and 10 represents the number of sensors. The 100 raw data points obtained from each sensor were converted into a 1010 matrix, and the 10 matrices of the sensors were concatenated into a 101010 matrix as a 10-channel input of the CNN backbone…”

We hope our response can remove your concerns and the revision could meet your requirements.

References:

  • Tarsha-Kurdi, F.; Landes, T.; Grussenmeyer, P.; Koehl, M. Model-driven and data-driven approaches using LIDAR data: Analysis and comparison. In ISPRS workshop, photogrammetric image analysis (PIA07). 2007, pp. 87-92.
  • Liu, P.; He, G.; Zhao, L. From Model-driven to Data-driven: A Survey on Active Deep Learning. 2021, arXiv preprint arXiv:2101.09933.

Reviewer#3, Concern # 5:

- Line 118, 124: standardize the notation of geographic coordinates

Author response: Dear reviewer, thanks a lot for your attentions and suggestions.

We have revised the notation of geographic coordinates in Section 2.1 General Information of Tea Samples (Line 124 and Line 130, Page 3, Clean revised version).

“…in Huangshan City, Anhui Province (29°42’33.26”N, 118°19’02.37”E)…”

“…in Xinyang City, Henan Province (32°07’23.79”N, 114°04’30.11”E)…”

We have improved the grammar and language by a 3rd party service for language polishing. Now, a deep revision of the manuscript is uploaded for your review.

According to your suggestions, now the readability of our manuscript is strongly increased. We really hope that we could have a deeper communication on science with you in the future. Thanks again for your supporting to our work.

Reviewer 4 Report

The manuscript concerns the use of commercial e-nose for tea classification. It is another publication among many on this topic. However, the key point in it is the use of a new algorithm for data reduction and classification. I believe that this is, unfortunately, the only strong point of this manuscript. I think that this article can only be published after a major revision process.

I have the biggest objection to the planning of the experiment. Why did the authors decide to run the measurement for 12 days? What relevance does this have in assessing the authenticity of the teas? Also, the authors did not state how many replicates of one type of tea they performed during their experiment. This would have been worth adding because currently, the reader does not know what the inter-group variation was. What was the procedure for cleaning the sensor chamber between measurements?

Introduction:

The authors often use simplifications that are unlikely to be used in scientific papers. Electronic noses do not detect aroma or bring any olfactory information. They only register the presence of volatile compounds in the gas mixture. So these may be compounds that do not affect the aroma at all (please refer to the http://dx.doi.org/10.1016/j.snb.2014.07.087).

Similarly in line 152: ‘allow the tea odour to disperse into…’. It isn’t the ‘odor’ disperse into sampler but volatile compounds.

Line 159: The sensor does not react with the gas, the reaction takes place on the surface of the active element of the sensor.

Table 2: I believe the term 'sensitive substance' is misleading and may confuse the reader

Line 233: Why do authors conclude that sensitivity to a specific substance leads to high cross-sensitivity? I don't see the connection.

Figure 10: Please convert sample labels to capital letters

Line 307: How did the authors deduce from the PCA that 'odor is rich and complex’?

Conclusion:

The sentences listed are not a conclusion but rather a summary. Please add information indicating further perspectives and how the presented results may influence the development of food authentication.

Author Response

Original Manuscript ID: foods-1096164     

Original Article Title: “A machine learning method for the fine-grained classification of green tea with geographical indication using a MOS-based electronic nose

Reviewer #4 (Comments and Suggestions for Authors):
The manuscript concerns the use of commercial e-nose for tea classification. It is another publication among many on this topic. However, the key point in it is the use of a new algorithm for data reduction and classification. I believe that this is, unfortunately, the only strong point of this manuscript. I think that this article can only be published after a major revision process.

I have the biggest objection to the planning of the experiment. Why did the authors decide to run the measurement for 12 days? What relevance does this have in assessing the authenticity of the teas? Also, the authors did not state how many replicates of one type of tea they performed during their experiment. This would have been worth adding because currently, the reader does not know what the inter-group variation was. What was the procedure for cleaning the sensor chamber between measurements?

Introduction:

The authors often use simplifications that are unlikely to be used in scientific papers. Electronic noses do not detect aroma or bring any olfactory information. They only register the presence of volatile compounds in the gas mixture. So these may be compounds that do not affect the aroma at all (please refer to the http://dx.doi.org/10.1016/j.snb.2014.07.087).

Similarly in line 152: ‘allow the tea odour to disperse into…’. It isn’t the ‘odor’ disperse into sampler but volatile compounds.

Line 159: The sensor does not react with the gas, the reaction takes place on the surface of the active element of the sensor.

Table 2: I believe the term 'sensitive substance' is misleading and may confuse the reader

Line 233: Why do authors conclude that sensitivity to a specific substance leads to high cross-sensitivity? I don't see the connection.

Figure 10: Please convert sample labels to capital letters

Line 307: How did the authors deduce from the PCA that 'odor is rich and complex’?

Conclusion:

The sentences listed are not a conclusion but rather a summary. Please add information indicating further perspectives and how the presented results may influence the development of food authentication.

Dear reviewer,

Thanks for your supporting. We appreciate that you give us such constructive advices and a literature with novel insights on E-nose. We have carefully read the literature and updated our basic knowledge. According to your suggestions, we have carefully revised our manuscript. Our responses for your questions are as follows:

Reviewer #4, Concern # 1:

- I have the biggest objection to the planning of the experiment. Why did the authors decide to run the measurement for 12 days? What relevance does this have in assessing the authenticity of the teas? Also, the authors did not state how many replicates of one type of tea they performed during their experiment. This would have been worth adding because currently, the reader does not know what the inter-group variation was. What was the procedure for cleaning the sensor chamber between measurements?

Author response: Dear reviewer, thanks a lot for your attentions and suggestions.

1. Thanks for your attentions to the planning of the experiments in our study. We are so sorry that making your confusion on the planning of our experiments. We would like to make some explanations as follows:

In this study, our main aim is proposing a machine learning method for the fine-grained classification of green teas by means of the MOS-based E-nose. As far as we known, there are two principal approaches of building machine learning models: the model-driven (parametric) approach and the data-driven (non-parametric) approach [1]. The model-driven approach needs to design selectors (strategy or model to select samples from candidate data set) with parameter by handcraft feature or metric. For the data-driven approach, their selectors adopted deep architecture whose features are automatically generated but not by handcraft [2]. In our study, PCA, an unsupervised method, was first used to distinguish the 12 green teas. As shown in Figure 10 (Figure10. Projection of the first two principal components of the PCA of the tea samples, Page 6, Clean revised version), the PCA results show a relatively high overlap between the classes, indicating that the PCA is not suitable for separating the classes. We proposed a novel framework (a data-driven model) consisting of a convolutional neural network backbone (CNN backbone) and a support vector machine classifier (SVM classifier), namely, CNN-SVM. In the framework, the deep CNN is designed to automatically mine the volatile compound information of the tea samples from E-nose data. The CNN can mine the potential information by automatically extracting local and global features and feed the information to the classifier providing a good performance. So, for the CNN-SVM (a data-driven model), the quantity and quality of data is important for the model’s good performance. Overfitting is a fundamental issue in data-driven models which prevents us from perfectly generalizing the models to well fit observed data on training data, as well as unseen data on testing set. Because of existence of overfitting, the model performs perfectly on training set, while fitting poorly on testing set. This is due to that over-fitted model has difficulty copying with pieces of the information in the testing set, which may be different from those in the training set. The causes of this phenomenon might be the noise learning on the training set: when the training set is too small in size (measuring during a short period of time), or has less representative data (measuring the same samples repeatedly) [3].

As you said, the experiments were conducted for 12 days in our study. For this planning of experiments, we have some thoughts as follows:

a. As is introduced in the shared literature [4], measurement uncertainty is one of the information losses connected with gas sensors, and gas sensor drift has a rather complex and inevitable effect for the sensor’s measurement properties. The time drift of gas sensor consists of a random temporal variation of the sensor response when it is exposed to the same analytes under identical conditions. The drift could make initial traditional machine learning method unsatisfactory for gas recognition after a relatively short period (typically few weeks) [5]. Frequent recalibrations are needed for to preserve accuracy. Therefore, in order to verify the robustness of our proposed method which could overcome the negative effect of time drift (to an extent), we chose to conduct a set of experiments in 12 days. The data in the first nine days was used for training and that in the last three days was used for testing. The experimental results indicated that our proposed model presented good performances and strong robustness for the fine-grained classification of the 12 teas, which shows a potential to avoiding the negative effect of the time drift in E-nose.

b. As we introduced before, the data-driven model’s performance can be significantly affected by the quantity and quality of dataset. Model training is actually a process of tuning its hyper-parameter. Well-tuned parameters make a good balance between training accuracy and regularity, and then inhibit the effect of overfitting. To tune these parameters, the model needs sufficient samples for learning. Therefore, in order to reduce the effects of overfitting, we chose to conduct the experiments for 12 days to obtain more training data. The tea samples for daily experiments were updated to obtain more representative data. The data in the first nine days was used for training and that in the last three days was used for testing. The experimental results indicated that our proposed model achieved a good performance on both the training and testing set.

c. Generally, a series of measurements conducted under the same conditions is called an equal-precision measurement [6]. Strictly speaking, there is no equal-precision measurement in the practical applications of E-nose. In our study, the experiments were conducted for 12 days in a clean testing room of the author’ laboratory (with good ventilation and an area of about 45 square meters) at a temperature of 25 °C ± 1 °C and a humidity level of 40% ± 2%. All the experiments were performed by a professional and experienced operator using the same devices. So, we think that there is no gross error in our measurements. The 12 green teas (six Maofeng teas and six Maojian teas) were measured 10 times per day and the tea samples for daily experiments were updated. So, the measurements conducted on different days were independent and could be regarded as unequal-precision measurements. These independent measurements satisfy the conditions of E-nose applications in real scenarios and improve the reliability and authenticity of the dataset, which can be used to verify the generalization of proposed method. In addition, as we know, the calculation of fusion weight is essential in the unequal-precision measurement, which limits the data process precision by traditional statistical method [7]. Our proposed method does not have to calculate the weights, and has good performances for the classification of green teas.

2. Thanks for your attentions to the number of replicates of one type of tea performed during our experiment. As we introduced in Section 2.2 Experimental Samples and Conditions (Line 149 and 176, Page 4, Clean revised version), the measurement of each type of tea was repeated 10 times per day during our experiments. Please check it carefully.

“…The 12 green teas (six Maofeng teas and six Maojian teas) were measured 10 times per day by a professional and experienced operator... Therefore, the dataset contained 1440 measurements (12 tea samples 10 measuring times 12 days).”

3. Thanks for your attentions to the procedure for cleaning the sensor chamber between measurements. We are so sorry that we did not introduce the details of the cleaning process for the sensor chamber between measurements.

The PEN3 (produced by “Airsense Analytics GmbH, Germany”), a commercial instrument with 10-channel MOS-based sensors, was used in our study. Because the PEN3 is a well-known commercial MOS-based E-nose (more details can be found on the website of the PEN3, www.airsense.com), we did not describe more details in our manuscript. As you pointed out, the cleaning process for the sensor chamber between measurements is important in the experiments. We would like to make some explanations as follows:

a. First at all, we added more details of our experiment conditions and the description of the reference air in our experiments in Section 2.2 Experimental Samples and Conditions (Line 147-155, Page 4, Clean revised version).

“The experiments were conducted for 12 days in a clean testing room of the authors' laboratory (with good ventilation and an area of about 45 square meters) at a temperature of 25 °C ± 1 °C and a humidity level of 40% ± 2%. The 12 green teas (six Maofeng teas and six Maojian teas) were measured 10 times per day by a professional and experienced operator, and the tea samples for daily experiments were updated. The acquisition of the volatile compound profile was conducted in a well-ventilated location to minimize baseline fluctuations and interference from other volatile compounds. The zero gas (a baseline) was produced using two active charcoal filters (Filter 1 and Filter 2 in Fig. 4) to ensure that the reference air and the air used for the samples had the same source.”

b. According to the advices of the PEN3 E-Nose producer, the active charcoal filters could be exchanged after about 1500 times of use. We have not reached this usage yet, so the baseline was stable in our experiments. The effectiveness of an individual filter was determined by cross-checking the two filters in a zero-run, where zero gas is subsequently produced in both filters and run over the sensors.

c. We have revised the Figure 4 (Figure4. Schematic diagram of the PEN-3 workflow, Page 7, Clean revised version) and added more details about the gas flow diagram of the PEN3 E-nose. In our experiments, the operation principle was conducted as follows:

  • For monitoring, an internal pump (pump 1, in the revised Fig. 4) is sucking the sample gas compounds through the sensor array.
  • For dilution, a second pump (pump 2, in the revised Fig. 4) is transferring filtered reference air into the sensor array. This clean air is also used to rinse the system.
  • The working environment air filtered with gas filter containing active charcoal is called zero gas. By using the zero-gas and comparing it to the signals from the analyzed sample gas the effect of the possible drift of the sensor itself is reduced (differential measuring technique). With the zero-gas the sensor array is kept clean to achieve a long lifetime.
  • As shown in the revised Fig. 4, the zero gas (a baseline) was produced from the working environment gas via two active charcoal filters (filter1 and filter2 in the revised Fig. 4) to ensure the reference air and the air used for the samples were from the same source.
  • Before the measurement, clean air was pumped through filter 2 into the E-nose with a flow rate of 10 ml/s in the direction of In 2 for 100 s. The automatic adjustment and calibration conducted by the zero gas is also called zero-point trim, in which the values relative to the zero-point values are recorded as a baseline.
  • In the collection stage, the tea sample's volatile gas in the sampler was pumped into the E-nose with a flow rate of 10 ml/s in the direction of In 1 to contact the sensor array for 100 s.
  • The flushing stage was conducted for cleaning the sensor chamber using the zero gas between measurements. In this step, the clean air was pumped into the E-nose with a flow rate of 10ml/s in the direction of In 2 for 100s. The zero gas flushed the sensor surface to completely remove the analytes.

d. According to your suggestions, we have added more details of the cleaning process for the sensor chamber between measurements in Section 2.2 Experimental Samples and Conditions (Line 156-177, Page 4, Clean revised version).

“The workflow of the E-nose includes the collection stage and flushing stage. As is shown in Fig. 4, for each tea, a 4 g tea sample was placed into a 50 ml sampler for 180 s to allow the tea’s volatile compounds to disperse into the sampler. Before the measurement, clean air was pumped through filter 2 into the E-nose with a flow rate of 10 ml/s in the direction of In 2 for 100 s. The automatic adjustment and calibration of the zero gas is called zero-point trim; the values relative to the zero-point values were recorded as a baseline. After the calibration, the tea sample's volatile gas in the sampler was pumped into the E-nose with a flow rate of 10 ml/s in the direction of In 1 to contact the sensor array for 100 s. The gas molecules were adsorbed on the sensors' surface, changing the sensors' conductivity due to the redox reaction on the surface of the sensor’s active element. The sensors' conductivity eventually stabilized at a constant value when the adsorption was saturated. The collection stage lasted 100 s, and sampling continued at one sample per second. Figure 5 shows examples of the 10 sensors' response curves in the collection stage for the 12 tea samples. The response value (R) of the sensor was calculated with the equation R = G0/G, where G0 is the conductivity of the sensor in reference air (in the calibration stage), and G is the conductivity of the sensor exposed to the sample vapor (in the data collection stage). The sensor chamber was flushed with the zero gas between measurements. In this step, the clean air was pumped into the E-nose with a flow rate of 10ml/s in the direction of In 2 for 100s. The zero gas flushed the sensor surface to completely remove the analytes. The flushing and data collection stages were repeated to obtain the raw data of 12 tea samples. Therefore, the dataset contained 1440 measurements (12 tea samples 10 measuring times 12 days).”

We hope our responses can remove your concerns and the revised version can meet your requirements.

Reference:

  • Tarsha-Kurdi, F., Landes, T., Grussenmeyer, P., & Koehl, M. (2007, September). Model-driven and data-driven approaches using LIDAR data: Analysis and comparison. In ISPRS workshop, photogrammetric image analysis (PIA07) (pp. 87-92).
  • Liu, P., He, G., & Zhao, L. (2021). From Model-driven to Data-driven: A Survey on Active Deep Learning. arXiv preprint arXiv:2101.09933.
  • Ying, X. (2019, February). An overview of overfitting and its solutions. In Journal of Physics: Conference Series (Vol. 1168, No. 2, p. 022022). IOP Publishing.
  • Boeker, P. (2014). On ‘electronic nose’methodology. Sensors and Actuators B: Chemical, 204, 2-17.
  • Padilla, M., Perera, A., Montoliu, I., Chaudry, A., Persaud, K., & Marco, S. (2010). Drift compensation of gas sensor array data by orthogonal signal correction. Chemometrics and Intelligent Laboratory Systems, 100(1), 28-35.
  • Zheng, H., & Gu, Y. (2021). EnCNN-UPMWS: Waste Classification by a CNN Ensemble Using the UPM Weighting Strategy. Electronics, 10(4), 427.
  • Wang, J., He, Z., Zhou, H., Li, S., & Zhou, X. (2017). Optimal weight and parameter estimation of multi-structure and unequal-precision data fusion. Chinese Journal of Electronics, 26(6), 1245-1253.

Reviewer#4, Concern # 2:

- Introduction: The authors often use simplifications that are unlikely to be used in scientific papers. Electronic noses do not detect aroma or bring any olfactory information. They only register the presence of volatile compounds in the gas mixture. So these may be compounds that do not affect the aroma at all (please refer to the http://dx.doi.org/10.1016/j.snb.2014.07.087).

Author response: Dear reviewer, thanks a lot for your attentions and suggestions.

1. Thanks for your attentions to the simplifications in the Section “Introduction” in our manuscript. We have added all the abbreviations used in our manuscript in the Section Abbreviations (Line 457, Page 16, Clean revised version). The details of the abbreviations are partly showed as follows (in the Section “Introduction”):

GTSGI

Green Tea Sub-categories with Geographical Indications

FGTSGI

Famous Green Tea Sub-categories with Geographical Indications

E-nose

Electronic Nose

CNN

Convolutional Neural Network

MOS

Metal Oxide Semiconductor

SVM

Support Vector Machine

The six abbreviations are all defined in parentheses the first time they appear in the abstract and main text, and are used consistently thereafter. As you said, the “GTSGI” and “FGTSGI” are defined by ourselves and the other four abbreviations are often used in scientific papers. The “GTSGI” and “FGTSGI” are two important objects in our manuscript. In our study, we proposed a machine learning model for the classification of the GTSGI and the identification of the FGTSGI. The descriptions of the GTSGI and FGTSGI are shown as follows: Chinese green tea is known for its health-functional properties. There are many green tea categories, which have sub-categories with geographical indications (GTSGI). Several high-quality GTSGI planted in specific areas are labeled as famous GTSGI (FGTSGI) and are expensive. In our study, the proposed CNN-SVM model was verified in the twelve green tea sub-categories (belong to two main green tea categories). Our proposed CNN-SVM model presents good performances and strong robustness for the fine-grained classification of the twelve GTSGI. Furthermore, extra experiments, identifying HSMF (the most famous and expensive Maofeng green tea sub-category) and XYMJ (the most famous and expensive Maojian green tea sub-category) were performed. The identification results for the two FGTSGI show the CNN-SVM achieved the highest F1 scores (97.77% for HSMF and 99.31% for XYMJ) among the five machine learning models. Therefore, we would like to use the two abbreviations in our manuscript to show the aims of our study more clearly. Thanks for your understanding.

2. Thanks a lot for pointing out the better fitting terms for E-nose and sharing a scientific reference to us. We have carefully read the literature and updated our basic knowledge. We have revised the description in our manuscript and added the reference according to your suggestions.

a. Section 1 Introduction (Line 45-49, Page 2, Clean revised version)

“An electronic nose (E-nose) is a chemical measurement system that measures chemical properties of sample gases and consists of a sampling system, a sensor array unit, and a data acquisition and processing system [1]. The sensor array can detect volatile compounds in the sample gases using a sampling system and provides feature information (sensor responses in the array)…”

b. Section 1 Introduction (Line 84-86, Page 2, Clean revised version)

“…In 2019, Shi et al. employed a CNN model to extract the volatile compound profile of beer using an E-nose...”

c. Section 1 Introduction (Line 105-106, Page 3, Clean revised version)

“…In the framework, the deep CNN is designed to mine the volatile compound information of the tea samples…”

d. Section 2.2 Experimental Samples and Conditions (Line 143-145, Page 4, Clean revised version)

“…A commercial E-nose (PEN3, Airsense Analytics GmbH, Germany; the details of the PEN3 can be found on the company’s website), with 10 MOS sensors was used to acquire the volatile compound profile of the tea samples...”

e. Section 3 Proposed Method (Line 236-237, Page 9, Clean revised version)

“…The deep CNN structure was used to mine the volatile compound information of the tea from the raw data of the E-nose…”

The rigor of our revised manuscript is increased by means of your suggestions. Thanks so much. And we hope the revision could meet your requirements.

References:

  • Boeker, P. (2014). On ‘electronic nose’methodology. Sensors and Actuators B: Chemical, 204, 2-17.

Reviewer#4, Concern # 3:

- Similarly in line 152: ‘allow the tea odour to disperse into…’. It isn’t the ‘odor’ disperse into sampler but volatile compounds.

Author response: Dear reviewer, thanks a lot for your attentions and suggestions.

Thanks a lot for pointing out the better fitting term here. We have replaced the “odor” with the “volatile compounds” in Section 2.2 Experimental Samples and Conditions (Line 156-158, Page 4, Clean revised version).

“…As is shown in Fig. 4, for each tea, a 4 g tea sample was placed into a 50 ml sampler for 180 s to allow the tea’s volatile compounds to disperse into the sampler…”

The rigor of our revised manuscript is increased by means of your suggestions. Thanks so much. And we hope the revision could meet your requirements.

Reviewer#4, Concern # 4:

- Line 159: The sensor does not react with the gas, the reaction takes place on the surface of the active element of the sensor.

Author response: Dear reviewer, thanks a lot for your attentions and suggestions.

Thanks a lot for pointing out the better fitting description here. According to your suggestions, we have revised the description in Section 2.2 Experimental Samples and Conditions (Line 164-166, Page 4, Clean revised version).

“…The gas molecules were adsorbed on the sensors' surface, changing the sensors' conductivity due to the redox reaction on the surface of the sensor’s active element...”

The rigor of our revised manuscript is increased by means of your suggestions. Thanks so much. And we hope the revision could meet your requirements.

Reviewer#4, Concern # 5:

- Table 2: I believe the term 'sensitive substance' is misleading and may confuse the reader

Author response: Dear reviewer, thanks a lot for your attentions and suggestions.

The PEN3 used in our study is a commercial instrument (produced by “Airsense Analytics GmbH, Germany”; the details of PEN3 can be found on the company’s website, www.airsense.com) with 10-channel MOS-based sensors. According to your suggestions, we have replaced the “sensitive substance” with “main performance” and added more details in Table 2 (Table 2. Details of the 10 sensors in the PEN3, Page 6, Clean revised version).

The rigor of our revised manuscript is increased by means of your suggestions. Thanks so much. And we hope the revision could meet your requirements.

Reviewer#4, Concern # 6:

- Line 233: Why do authors conclude that sensitivity to a specific substance leads to high cross-sensitivity? I don't see the connection.

Author response: Dear reviewer, thanks a lot for your attentions.

We are so sorry that make your misunderstanding due to our unclear description. As is shown in Table 2 (Table 2. Details of the 10 sensors in the PEN3, Page 6, Clean revised version), each sensor in the E-nose is typically not sensitive for only a single kind of substance, but rather for a certain range of substances, some of which do overlap. Therefore, we think that it is essential to consider the correlation between the sensors and the sensor array in the E-nose has high cross-sensitivity. We have revised the description in our manuscript.

1. Section 1 Introduction (Line 90-93, Page 3, Clean revised version).

“…The sensor array in an E-nose system has high cross-sensitivity because each sensor in the E-nose is typically not sensitive for only a single kind of substance, but rather for a certain range of substances, some of which do overlap...”

2. Section 3.1 Data Preprocessing (Line 242-243, Page 9, Clean revised version).

“The 10-channel sensor array in the PEN3 system has high cross-sensitivity because each sensor is sensitive to a certain range of substances, and there is some overlap...”

The rigor of our revised manuscript is increased by means of your suggestions. Thanks so much. And we hope the revision could meet your requirements.

Reviewer#4, Concern # 7:

- Figure 10: Please convert sample labels to capital letters

Author response: Dear reviewer, thanks a lot for your attentions and suggestions.

According to your suggestions, we have converted the sample labels to capital letters in Figure 10 (Figure10. Projection of the first two principal components of the PCA of the tea samples, Page 12, Clean revised version).

The readability of our revised figure is increased by means of your suggestions. Thanks so much. And we hope the revision could meet your requirements.

Reviewer#4, Concern # 8:

- Line 307: How did the authors deduce from the PCA that 'odor is rich and complex’?

Author response: Dear reviewer, thanks a lot for your attentions.

We are so sorry that making your confusion due to our incorrect statement here. In our study, we want to use the PCA to distinguish the 12 green teas and prove that the PCA is ineffective for classifying these multiple highly similar green teas. We would like to make some explanations to you.

PCA is an unsupervised method commonly used for pattern recognition, which allows data to be grouped according to the similarity of the input characteristics to determine the distance between classes. If the clusters are well separated, high classification accuracy is expected. If the clusters are in close proximity or overlap, low accuracy is expected. As shown in Figure 10 (Figure10. Projection of the first two principal components of the PCA of the tea samples, Page 12, Clean revised version), the PCA results show a relatively high overlap between the classes, indicating that this method is not suitable for separating the classes. We have revised the description in our manuscript, and the specific revisions are as follows:

1. Section 4.1 Principal Component Analysis (Line 312-319, Page 12, Clean revised version).

“…The cumulative variance of PC 1 and PC 2 was 95.43%, showing a small loss of information. The PCA method allows data to be grouped according to the similarity of the input characteristics to determine the distance between classes as shown in Fig. 10. If the clusters are well separated, high classification accuracy is expected. If the clusters are in close proximity or overlap, low accuracy is expected. As shown in Fig. 10, the PCA results show a relatively high overlap between the classes, indicating that this method is not suitable for separating the classes.”

2. Section 5 Conclusions (Line 411-414, Page 15, Clean revised version)

“PCA, an unsupervised method, was used to show the separability of the 12 tea sample data (stable values) obtained from the E-nose. Not surprisingly, the PCA results showed that this method was ineffective for classifying the sample due to relatively high overlap...”

The rigor of our revised manuscript is increased by means of your suggestions. Thanks so much. And we hope the revision could meet your requirements.

Reviewer#4, Concern # 9:

- Conclusion: The sentences listed are not a conclusion but rather a summary. Please add information indicating further perspectives and how the presented results may influence the development of food authentication.

Author response: Dear reviewer, thanks a lot for your attentions and suggestions.

According to your suggestions, we have revised our Conclusion to add more information indicating further perspectives and the active influence for the development of food authentication in Section 5 Conclusions (Line 440-447, Page 15, Clean revised version).

“…In conclusion, the combined strategy of CNN and SVM enhanced the detection performance of multiple highly similar green teas. The proposed method provided high classification accuracy, showing that the tea quality differed for different geographical indications. Moreover, the method is rapid, convenient, and effective for classifying green teas. In the future, the potential of this method will be explored for other teas (or other foods) to expand the application of the proposed framework combined with the E-nose. We expect that the framework has a promising potential for using machine learning methods for food authentication…”

According to your suggestions, now the conclusion of our study is presented more clearly, and the readability of our manuscript is strongly increased.

We have improved the grammar and language by a 3rd party service for language polishing. Now, a deep revision of the manuscript is uploaded for your review.

We really hope that we could have a deeper communication on science with you in the future. Thanks again for all the very professional advices.

Round 2

Reviewer 2 Report

I agree that this paper is of interest in terms of methodology, but scientifically it is very basic with no novelty. There is no scientific discussion. The authors focused only on the methodology with several irrelevant figures including the Table 6. 

Author Response

Dear reviewer,

Thanks for your supporting. Our answers for your questions are as follows:

Author response: Dear reviewer, thanks for your attentions.

We are very glad to receive the responses from you after our revised version in Round 1. The readability of our manuscript is strongly increased according to your suggestions in Round 1. But we are very sorry that making you misunderstanding on the aims and novelty of our study. Anyway, we would like to make some explanations to you again.

1. As shown in the introduction of Foods journal, Foods is an international, peer-reviewed scientific open access journal which provides an advanced forum for studies related to all aspects of food research. The aim of the journal is to encourage professionals to publish their experimental and theoretical results. Our study proposed a machine learning method to identify green teas using a commercial E-nose for developing the applications of Artificial Intelligence (AI) technology in the area of “sensory and food quality” (from the scope list of Foods, https://www.mdpi.com/journal/foods/about).

2. In the title of our manuscript, we clearly show the aims of our study. From the title, there are two aims as shown in follows:

a. We proposed a machine learning framework (named CNN-SVM) which is consisted of a CNN backbone and a SVM classifier.

b. We presented the fine-grained classification of green teas with geographical indication by means of the proposed CNN-SVM framework. The data of the teas’ samples were collected from a commercial MOS-based E-nose (called PEN3).

3. In this study, we proposed a machine learning model for the classification of the GTSGI and the identification of the FGTSGI. The descriptions of the GTSGI and FGTSGI are shown as follows: Chinese green tea is known for its health-functional properties. There are many green tea categories, which have sub-categories with geographical indications (GTSGI). Several high-quality GTSGI planted in specific areas are labeled as famous GTSGI (FGTSGI) and are expensive. In our study, the proposed CNN-SVM model was verified in the twelve green tea sub-categories (belong to two main green tea categories). Our proposed CNN-SVM model presents good performances and strong robustness for the fine-grained classification of the twelve GTSGI. Furthermore, extra experiments, identifying HSMF (the most famous and expensive Maofeng green tea sub-category) and XYMJ (the most famous and expensive Maojian green tea sub-category) were performed. The identification results for the two FGTSGI show the CNN-SVM achieved the highest F1 scores (97.77% for HSMF and 99.31% for XYMJ) among the five machine learning models.

4. As far as we known, there are many studies that primarily focused on distinguishing teas in different categories and teas in the same category from different production areas or with different qualities, respectively. Few studies have focused on the fine-grained classification of tea in different categories (categories) and from different production areas (sub-categories) simultaneously. Green teas usually have light aromas and these aromas have small differences among different green tea categories. An increasing number of sub-categories has complicated the classification of these similar samples, because there are not only tiny differences in green teas among categories but also much tinier differences among sub-categories. In our study, we used two main green teas’ categories (Maofeng tea category and Maojian tea category). Each category contained six sub-categories (six Maofeng tea sub-categories and six Maojian tea sub-categories), respectively. All the tea samples were picked and processed 14 days before the Pure Brightness Festival (around April 5 or 6) and were called “before-brightness tea”. As we introduced in our manuscript, the twelve green teas sub-categories, belong to two categories, not only have a high similarity in the aroma among the sub-categories, but also have tiny differences between the two categories. We proposed a machine learning method, named CNN-SVM, for the fine-grained classification of green tea with geographical indication using an electronic nose. In our work, we analyzed the network structure of the proposed CNN-SVM and verified the effectiveness of the CNN-SVM via the twelve green teas. Furthermore, we presented the good performance of our proposed CNN-SVM for the identification of the two most famous and expensive teas with protected designation of origin (HSMF and XYMJ) via extra experiments. The proposed framework had the best performance for both the classification of the GTSGI and the identification of the FGTSGI.

5. The effectiveness of our proposed CNN-SVM model was demonstrated on the classification of the twelve green teas with high accuracy and strong robustness. As we introduced before, green teas usually have light aromas which are much weaker than those of the other common tea categories in China, such as oolong tea, black tea, and jasmine tea. The study provides a novel insight for distinguishing multiple highly similar green tea and might have a potential for other tea’s fine-grained classification tasks in the future.

6. In the Introduction Section of our manuscript, we have clearly shown the contributions of our study as follows:

a. A preprocessing scheme inspired by the multi-channel input to the CNN model used in image processing is proposed for the 10-channel E-nose’s data. The channel’s sensor data are converted into a single matrix, and the combined matrices of the 10 channels represent the multi-channel data.

b. The network structure of the proposed CNN-SVM, a CNN backbone with a SVM classifier, is analyzed. In the framework, the deep CNN is designed to mine the aroma features of the tea, and the SVM is used to classify the data (small sample sizes) to improve the classification performance.

c. A comprehensive study on the fine-grained classification of green tea with geographical indication is presented, demonstrating the high accuracy and strong robustness of the proposed CNN-SVM framework.

7. In the Conclusion Section of our manuscript, we have clearly shown the main conclusions as follows:

a. PCA was used to illustrate the data distribution of the 12 tea samples using stable values of the E-nose signals. The result indicated that the aroma features of the green tea were complex, and the differences (between the tea categories and sub-categories) were relatively small. The stable values combined with an SVM model were used to distinguish the teas; this approach had poor performance. The results showed that the static characteristics were not sufficient for the fine-grained classification of the 12 green teas, and meaningful feature information was lost.

b. A 10-channel input matrix, which was obtained by converting the raw data of the 10 MOS-based sensors (each sensor for one channel) in the E-nose system, was constructed to mine the deep features using the CNN model. The multi-channel design considered the cross-sensitivity of the sensors and contained sufficient details for the classification tasks, providing a novel feature extraction method for E-nose signals in practical applications.

c. A CNN-SVM framework consisting of the CNN backbone and the SVM classifier was proposed to perform the classification using the 10-channel input matrix. The novel structure of the CNN backbone based on ResNeXt was effective for extracting the deep features from the different channels automatically. The SVM classifier improved the generalization ability of the CNN model and increased the classification accuracy from 91.39% (CNN backbone + CNN classifier) to 96.11% (CNN backbone + SVM classifier) among the 12 green teas due to its good discrimination ability for small sample sizes.

d. Compared with the other four machine learning models (SVM, CNN-Shi, CNN-SVM-Shi, and CNN), the proposed CNN-SVM provided the highest scores of the five evaluation metrics for the classification of the GTSGI: accuracy of 96.11%, recall of 96.11%, precision of 96.86%, F1 score of 96.03%, and Kappa score of 95.76%. Excellent performance was obtained for identifying the FGTSGI, with the highest F1 scores of 97.77% (for HSMF) and 99.31% (for XYMJ). These experimental results demonstrated the effectiveness of the CNN-SVM for the classification of the GTSGI and the identification of the FGTSGI.

e. This study provides novel insights into fine-grained classification using an E-nose system, protecting consumers’ interests and producers’ reputations. The proposed method also shows a strong potential for distinguishing multiple highly similar green teas and may be applicable to the fine-grained classifications of other teas in the future.

8. About your confusion on Table 6 (“The authors focused only on the methodology with several irrelevant figures including the Table 6.”). We do not know where is the Table 6 in our manuscript (because there are only five Tables in total in our manuscript).

9. According to your suggestions, we have improved the grammar and language again by a 3rd party service for language polishing. Now, a revised version of our manuscript is uploaded for your review.

We really hope that our explanations could remove your concerns. Thanks for your understanding. We really hope that we could have a deeper communication on science with you in the future. Thanks again for your supporting to our work.

Reviewer 4 Report

Thank you. I don't have any additional questions.